# Designing pathways for bioproducing complex chemicals by combining tools for pathway extraction and ranking

Anastasia Sveshnikova [1,2], Omid Oftadeh [1,2] & Vassily Hatzimanikatis [1]

The synthesis of many important biochemicals involves complex molecules and numerous reactions. The design and optimization of whole-cell biocatalysts for the production of these molecules requires metabolic modeling to extract production pathways from biochemical databases and integrate them into genome-scale models of the host. However, the synthesis of such complex molecules often requires reactions from multiple pathways operating in balanced subnetworks that are not assembled in existing databases. Here, we present SubNetX, a computational algorithm that extracts reactions from a database and assembles balanced subnetworks to produce a target biochemical from selected precursor metabolites, energy currencies, and cofactors. These subnetworks can be integrated into whole-cell models, allowing the reconstruction and ranking of alternative biosynthetic pathways based on yield, length, and other design goals. We apply SubNetX to 70 industrially relevant natural and synthetic chemicals to demonstrate the application of this pipeline.

The green chemistry movement is shifting the production of important chemicals, such as pharmaceuticals or food additives[1], away from traditional fossil–fuel-based chemical syntheses and costly agricultural extraction towards bioproduction[2]. Eco-friendly and sustainable bioproduction uses microbes to produce chemicals of interest, either taking advantage of existing biosynthetic pathways or through metabolic engineering[3]. However, the complexity of biochemicals often limits their industrial scalability, and engineering strategies are currently limited to relatively simple compounds (e.g., ethanol and 1,3-butanol)[4,5]. One reason for this is that engineering strategies mainly propose linear pathways in contrast with living organisms that can combine metabolic pathways to generate complex secondary metabolites. Inspired by nature, researchers thus aim to design pathways that divert resources from several pathways toward a single target to achieve higher yields for complex natural and non-natural compounds.

Engineering strategies rely on biochemical databases to search and suggest pathways. Some biochemical databases contain the natural reactions observed in nature[6,7], while others contain computationally predicted reactions that attempt to represent the conceivable biochemical space[8,9]. Different computational algorithms are used to search these databases to find bioproduction pathways to a particular target compound. Currently, bioproduction pathways are designed using three classes of computational tools (for a comprehensive review, see Wang et al.[10]): (i) graph-based approaches that use graph-search algorithms to find pathways, (ii) stoichiometric approaches that use constraint-based optimization to find pathways, and (iii) retro-biosynthesis approaches that use algebraic operations to propose novel reactions, i.e., reactions not observed in nature. Graph-based and retrobiosynthesis methods both rely on graph-search algorithms, enabling them to navigate large networks of biochemical reactions. However, these pathways are a linear combination of heterologous reactions, limiting the output of such methods to pathways with a single precursor among the host metabolites[10]. However, a pathway is not stoichiometrically feasible if the required cosubstrates and cofactors are not connected to the host metabolism, a potential shortcoming of linear pathways. On the other hand, stoichiometric approaches allow for the analysis of subnetworks connected to the host metabolism via multiple precursors and their evaluation in the

[1]Laboratory of Computational Systems Biotechnology, École Polytechnique Fédérale de Lausanne (EPFL), Lausanne, Switzerland. [2]These authors contributed equally: Anastasia Sveshnikova, Omid Oftadeh. ✉e-mail: vassily.hatzimanikatis@epfl.ch

context of the host metabolism. As such, they often yield feasible pathways. However, constraint-based approaches are sensitive to the size of the reaction network due to limited computational power and cannot account for conceivable xenobiotic metabolism that contains hundreds of thousands of reactions[8,11–13].

Here, we address issues in existing pathway-design tools by combining the strengths of constraint-based and retrobiosynthesis methods into a pipeline, termed SubNetX (for Subnetwork extraction). The innovation of our algorithm lies in assembling a hypergraph-like network as an intermediate step in pathway design. This network defines a feasible solution space that connects a target molecule to the native metabolism of the host organism. Crucially, it integrates mechanistic details, including thermodynamics and kinetics, to enhance the reliability and precision of pathway predictions. This pipeline allows the exploration of large reaction networks to find an optimal pathway for the bioproduction of a target compound that would integrate into the native host metabolism, while accounting for the stoichiometric and thermodynamic feasibility of the pathways. SubNetX relies on constraint-based methods to ensure the feasibility and retrobiosynthesis methods to handle larger networks and identify biosynthetic pathways not observed in nature. We show that our approach predicts viable pathways for the synthesis of 70 chemical targets with higher production yields compared to linear pathways. The extracted subnetworks for these 70 compounds are available to the community as a resource for designing industrially valuable synthetic organisms. Furthermore, the pipeline allows the extraction of subnetworks for any other selected target compound that may be of interest. The pipeline is readily transferable to other biochemicals, and we discuss how it can be applied to other host organisms. Thus, we anticipate that SubNetX will be instrumental in designing pathways producing complex natural and non-natural compounds in various organisms.

## Results

### Subnetwork extraction towards the target compound

To maximize bioproduction strategies, we designed SubNetX to employ network exploration tools and constraint-based optimization methods to link required cosubstrates and subsequent byproducts to the host's native metabolism (Supplementary Fig. 1). The SubNetX workflow and its application to the prediction of balanced minimal subnetworks is illustrated in Fig. 1. The SubNetX workflow is organized into five main steps: (i) reaction network preparation where a database of elementally balanced reactions, target compounds, and precursor compounds are defined; (ii) graph search of linear core pathways from the precursor compounds to the target compounds, (iii) expansion and extraction of a balanced subnetwork where cosubstrates and byproducts are linked to the native metabolism, (iv) integration of the subnetwork into the host, and (v) ranking of the feasible pathways. Overall, the workflow requires as inputs: (i) a network of balanced biochemical reactions, i.e., a database, (ii) a set of target compounds, (iii) a set of precursors, which most commonly depend on the host of choice, (iv) a metabolic model of the host metabolism, and (v) user-defined parameters to adjust the search based on the user's needs. The network of balanced biochemical reactions can be either limited to known reactions or extended to include predicted biochemical reactions.

To find pathways to the targets and required cosubstrates in this work, we prepared a network of known and predicted biochemical reactions designed for the biosynthesis of aromatic compounds[13] (Fig. 1, steps 1–3). We then integrated the resulting subnetwork into the genome-scale metabolic model of *E. coli* (Fig. 1, step 4) to ensure that the target compound can be produced according to the metabolic capabilities of the host. As the extracted networks could contain thousands of reactions, it would be experimentally impossible to integrate the entire network into the host. Therefore, we used a mixed-integer linear programming (MILP) algorithm to identify sets of feasible pathways that will contain a much smaller number of heterologous steps (Fig. 1, step 5). We did this by finding the minimum number of essential reactions from the subnetwork that could produce the target compound, with each minimal set of reactions referred to as a feasible pathway. Finally, these feasible pathways were ranked based on yield, enzyme specificity, and thermodynamic feasibility (Fig. 1, step 5).

The inclusion of cheminformatics-predicted reactions allows SubNetX to identify novel pathways with potentially higher yields than those reported experimentally. By expanding the biochemical search space, these pathways offer innovative solutions for biosynthetic design while highlighting the necessity of experimental validation to address uncertainties in enzyme specificity and reaction mechanisms. Advances in structural modeling and validation tools, such as AlphaFold[14], can enhance the reliability of these predictions by assessing enzyme compatibility and reaction feasibility. This integration of cheminformatics and machine learning tools exemplifies the increasing feasibility of exploring and engineering hypothetical pathways in practical applications.

### Design of balanced pathways for complex secondary metabolites

As an initial test of SubNetX, we applied it to 70 metabolites used as pharmaceuticals (Supplementary Data 1). To ensure a representative and diverse selection of compounds for our study, we focused on the intersection of datasets from ARBRE and the Natural Product Atlas,

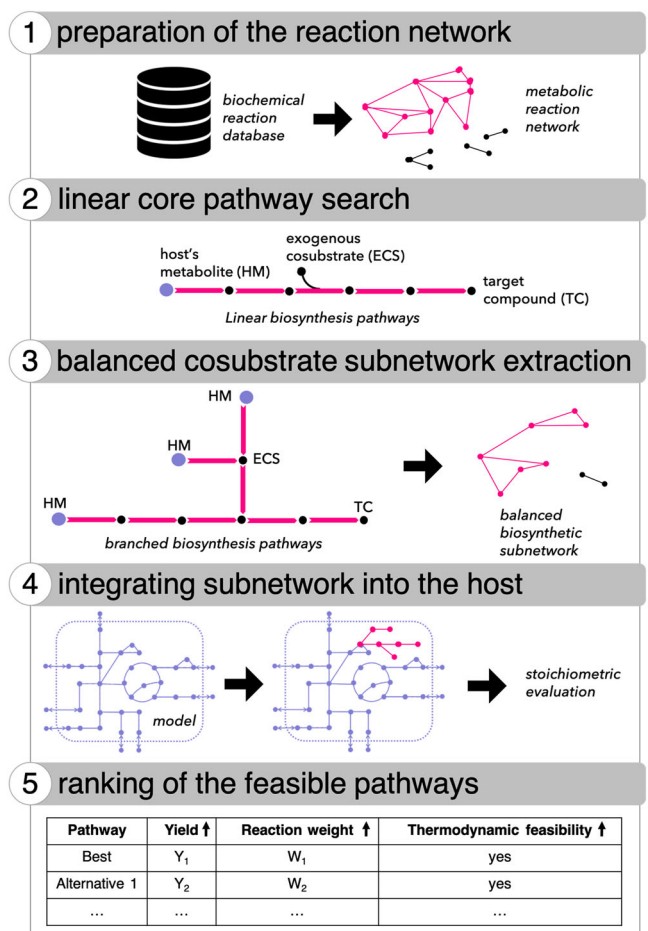

**Fig. 1 | The SubNetX workflow.** Step 1: preparation of the reaction network; step 2: linear core pathway search; step 3: balanced cosubstrate subnetwork extraction; step 4: integrating subnetwork into the host; and step 5: ranking feasible pathways.

incorporating structural and functional classifications from the ChEBI ontology to define drug-like compounds. From this curated dataset, we applied a randomized selection while preserving structural diversity. The selected compounds span a broad chemical space, ranging from small molecules such as β-nitropropanoate, azomycin, and DL-cycloserine (each containing only three carbon atoms) to larger, structurally complex metabolites such as β-carotene, which contains 40 carbon atoms. We evaluated the synthetic accessibility (SA) of a selected set of these compounds[15] (Supplementary Table 1) among a list of various natural compounds synthesized in different hosts (Supplementary Table 2) to represent the chemical diversity.

We searched the ARBRE network[13] to connect the target molecules to the host metabolism. ARBRE is a highly curated database of balanced biochemical reactions with a particular focus on industrially relevant aromatic compounds, comprising ~400,000 reactions. ATLASx, on the other hand, is one of the largest networks of predicted biochemical reactions, containing over 5 million reactions that span a wide range of biochemical space. Together, these databases serve as a comprehensive foundation, enabling SubNetX to process large-scale reaction networks and extract meaningful pathways efficiently. For each compound, we extracted a balanced subnetwork as described in "Methods" (Fig. 2).

Using only the ARBRE biochemical network, we mapped most of the target compounds to *E. coli* native metabolites. For the remaining compounds, we expanded the ARBRE network as needed. For example, for scopolamine, the ARBRE biochemical network did not contain the biosynthesis pathway, producing two tropane derivatives from putrescine, which is essential for scopolamine production (Supplementary Fig. 2). Thus, we supplemented these pathways using ATLASx[11], which can fill in missing reactions to connect scopolamine to the *E. coli* metabolism. ATLASx recovered a pathway to produce the two tropane derivatives already used to experimentally produce scopolamine in *E. coli*[16]. This pathway included one unbalanced reaction converting *N*-methylpyrrolinium to tropinone, which we replaced by two balanced reactions, chalcone synthase and tropinone synthase (Supplementary Fig. 2). These two reactions were annotated and added to ARBRE, which allowed us to create a balanced subnetwork for scopolamine. The synthesis of scopolamine provides an example of how potential gaps in biochemical knowledge can be identified and addressed while designing pathways to produce novel compounds.

Once we were able to extract subnetworks for all 70 compounds, we compared the sizes of the extracted subnetworks, finding them to be similar for different targets (Fig. 2). Interestingly, alternative feasible pathways required network expansion for the balanced synthesis of cofactors such as tetrahydrobiopterin that are only found in vertebrates[17]. Such non-native cofactors were only necessary for some pathways of the extracted subnetworks, while alternative feasible pathways existed involving only *E. coli* cofactors. Thus, to control network size and avoid unnecessary expansion, we recommend using a SubNetX search mode that avoids expansion around non-native cofactors.

## Constraint-based optimization extracts feasible branched pathways

Next, we evaluated the stoichiometric feasibility of the extracted subnetworks, which were designed to biosynthesize each compound using constraint-based optimization. To this end, we integrated the subnetwork for each compound into the *E. coli* genome-scale metabolic model iJO1366[18] and maximized the production of the target compounds. We found that for all 70 compounds, there was at least one feasible pathway to produce them with the maximum theoretical yield of 100% g-C, without loss of carbon to other byproducts. As in previous studies[3,19], we assumed glucose to be the limiting substrate by constraining its uptake (see "Methods"), while inorganic compounds were assumed to be available in excess. Notably, aside from the uptake of ammonia to provide nitrogen for the synthesis of nitrogen-

containing compounds, no uptake of other inorganics was observed in different alternative optimal pathways. This finding indicates that glucose uptake alone was sufficient to meet the chemical and redox costs required for target compound synthesis, including the production of ATP, NADH, and NADPH. In addition to the maximum yield, we determined the minimum number of heterologous reactions required to produce each compound at its maximum yield (Supplementary Data 1).

For most of the target compounds, there exist linear pathways, identified by the graph-search algorithms, that can be integrated into the host metabolism without the need for network expansion. We compared the product yield of such linear pathways with the product yield when SubNetX was used, and we found that higher yields could be achieved in the latter case. We analyzed the networks of the higher yield, and we found that they are operating as branched pathways of heterologous reactions that converge to product synthesis. These observations are illustrated via three pathway examples (Fig. 3). The first pathway (Fig. 3a) was linear and required four reactions to derive benzyl cinnamate from phenylalanine. As the second step of this pathway produced a byproduct not used in subsequent steps (i.e., glyoxylate), this carbon loss reduced the product yield. When we used the extracted network from SubNetX, a branched pathway to benzyl cinnamate (Fig. 3b) was derived from two native precursors, phenylalanine and 4-hydroxybenzoate. No byproducts were formed in this pathway, and all carbon atoms were conserved to produce the target. This demonstrates that SubNetX can provide feasible pathways, which can increase the product yield by reducing byproducts, which could even be toxic due to accumulation in the host.

The third example concerned the production of scopolamine. The shortest feasible pathway toward this compound (Fig. 3c) contained two separate branches of heterologous reactions to form tropate and tropine, each of which contained a part of scopolamine. Because different parts of its structure would have to be derived from different native precursors, there was no linear pathway of heterologous reactions to produce scopolamine. This example shows that SubNetX can extensively search far beyond the linear pathways obtained by the commonly used graph-search algorithms.

## Predicted pathways promise higher yields

Not all of the 70 target compounds in this study have been experimentally produced in *E. coli*. Therefore, we limited the comparison of computationally predicted and experimental pathways to a subset of eight compounds for which the experimental data were available. They are berberine, ajmalicine, scopolamine, strictosidine, *N*-cinnamoyl serotonin, benzyl benzoate, benzyl cinnamate, and quercetin 3-*O*-(6′-acetyl-glucoside). To compare the pathways predicted by SubNetX with those that have been implemented experimentally (Table 1), we first need to make the data comparable; experimentally implemented pathways are typically reported as a gene set (Supplementary Data 2–9), whereas the output of SubNetX is a reaction set. We, therefore, mapped the genes reported for the experimental pathways to their corresponding enzyme classes (EC) using UniProt (Supplementary Data 2–9) and mapped the predicted reactions of the pathways to their EC classes using BridgIT[20].

Then, for comparison, we looked at the first three or four levels of the EC numbers to match either the enzyme family or the exact enzyme. The pathways predicted by SubNetX retained some parts of the experimentally implemented pathways. For ajmalicine, 6 out of 12 reaction classes were shared with the experimentally implemented pathway, based on matching the first three levels of the EC numbers. Interestingly, 66 pathways predicted higher yields than those with the highest overlap (Table 1). It is noteworthy that although the exact reactions could be different, the predicted pathways passed through the same set of intermediates as the experimentally implemented pathways (e.g., 7-deoxyloganate, loganate, laganin, secologanin, and

 

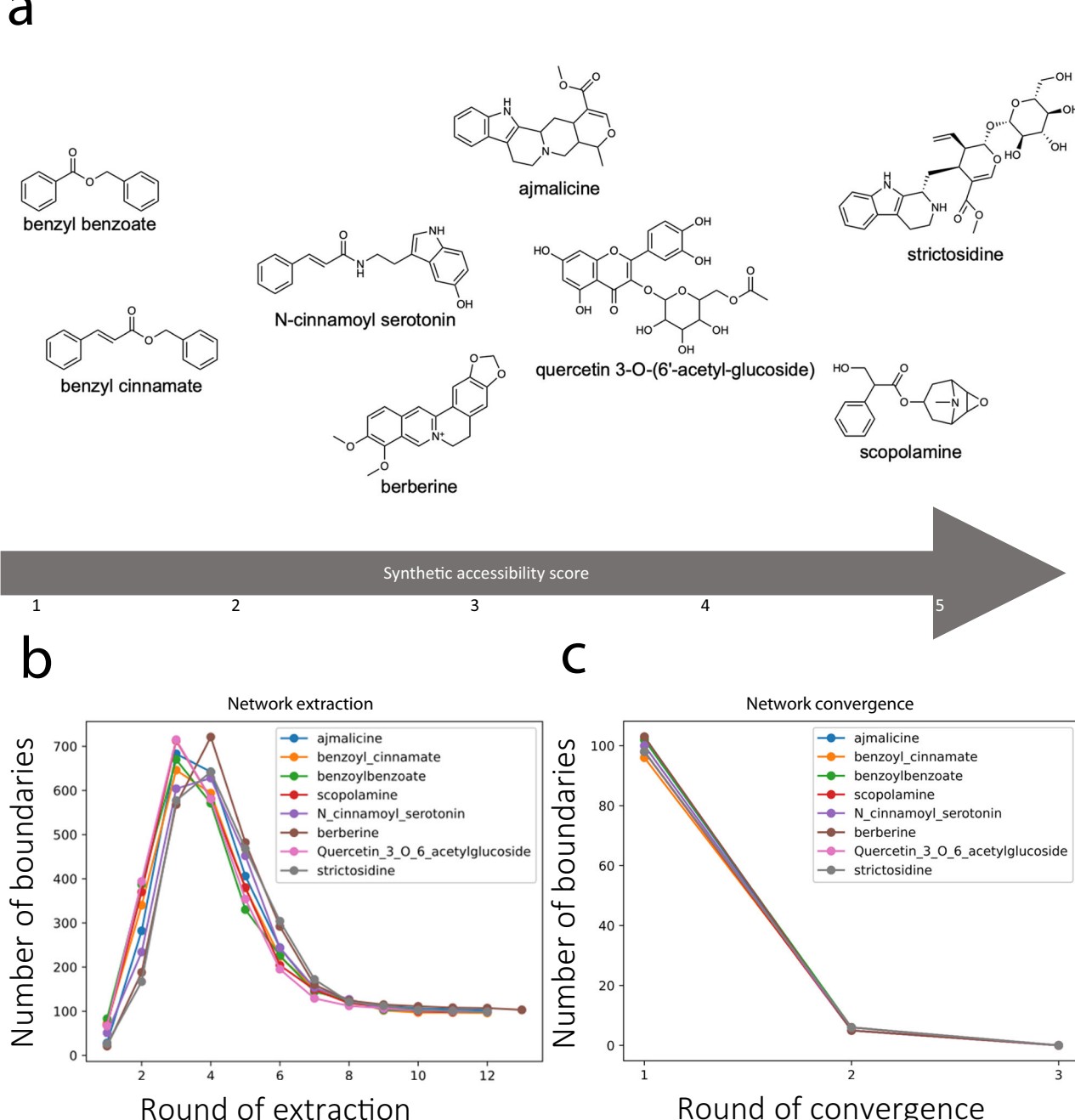

**Fig. 2 | Subnetwork extraction for a selected set of case studies. a** The chemical structures of selected compounds are shown alongside their synthetic accessibility scores, which evaluate how easily these compounds can be synthesized. **b** Network extraction: The number of boundaries (representing potential intermediate or limiting compounds) identified during each round of subnetwork extraction is plotted. This metric reflects the extent to which the reaction network expands, capturing the relevant metabolic context for predicting biosynthetic pathways. Peaks in this graph highlight the addition of important intermediates or branches to the network. **c** Network convergence: The progressive reduction in the number of boundaries across successive rounds of refinement is shown. This demonstrates how the algorithm systematically narrows the extended network by removing redundant or unlikely pathways, ultimately focusing on plausible biosynthetic routes.

strictosidine to produce ajmalicin), suggesting that SubNetX captured the essential biochemistry surrounding the experimentally implemented pathways. The difference in reaction steps was mainly due to the difference in cofactors and cosubstrates. This could have several reasons, including that the predicted pathways have been optimized for *E. coli*, whereas some experimental pathways were implemented in other organisms. Another reason could be that the experimentally implemented pathways might include reactions with unknown mechanisms or stoichiometries, which were not included in the model because the SubNetX search space is limited to a database of mass-balanced reactions with known stoichiometries. Once such reaction stoichiometries and mechanisms are known, they can be easily integrated into the SubNetX search space, e.g., ARBRE. Finally, this result demonstrates the ability of SubNetX to reconstruct pathways with higher yields, although this will need to be validated experimentally. Thus, despite limitations such as incomplete stoichiometric or mechanistic information and differences in annotation, we have demonstrated that SubNetX can retrieve a subset of experimentally

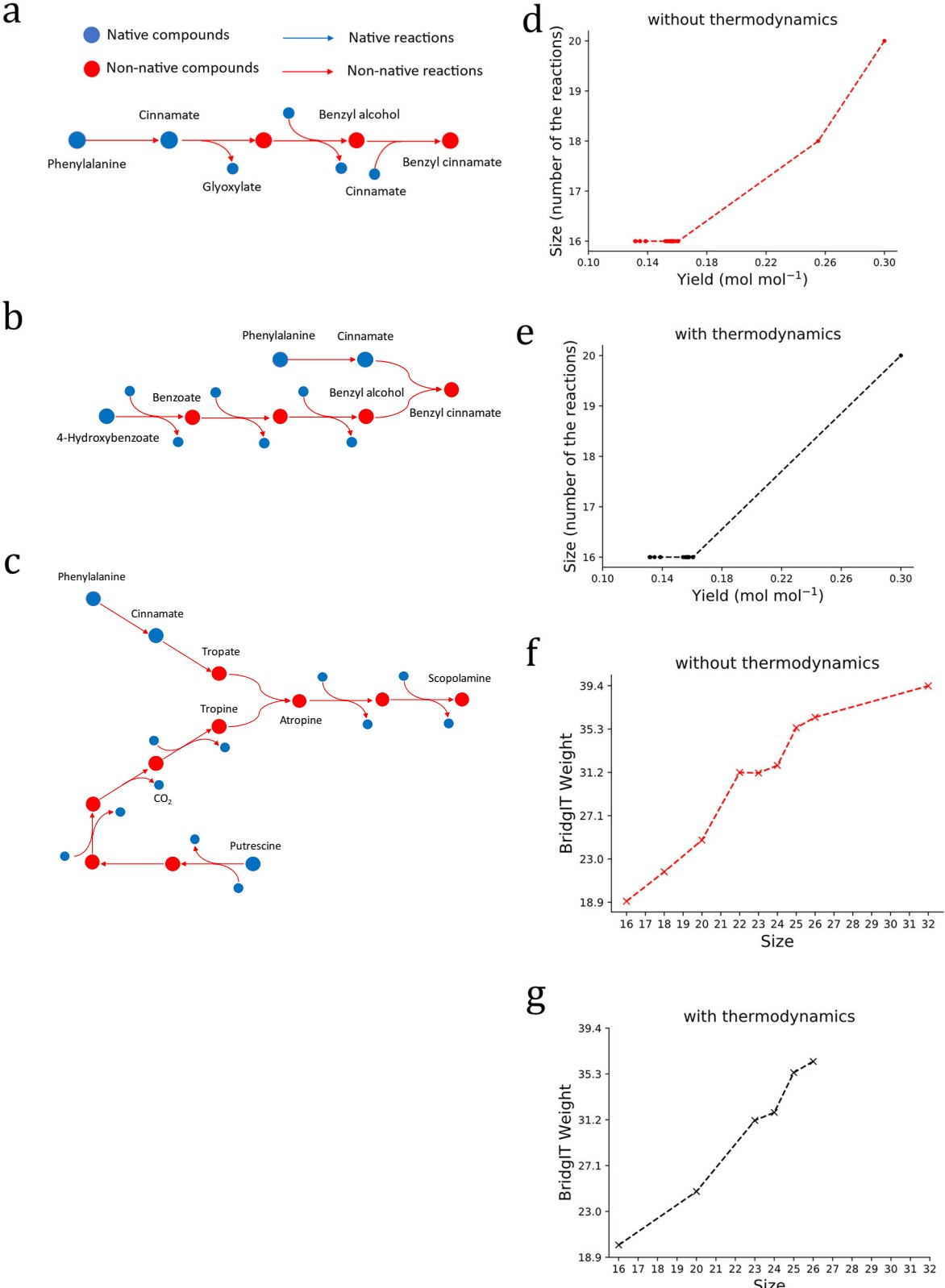

implemented pathways while suggesting potentially more efficient alternatives for the remaining pathways.

**Ranking the predicted pathways**
Next, for the above eight compounds, we sorted the pathways by assigning them a weight either by length or enzyme specificity using the MILP formulation. To weight pathways by length, we assigned

equal weights to all reactions. To weigh the pathways based on the likelihood of a correct enzyme assignment, we used the BridgIT score[20] (see "Methods") to weight the reactions. Here, if the pathway contains novel reactions, a lower weight reflects a higher likelihood that promiscuous enzymes exist to catalyze these novel steps[20]. Our goal was to investigate the tradeoffs between different engineering objectives, such as higher yield and shorter pathway length, while analyzing the

**Fig. 3 | Examples of feasible pathways and exploring the trade-off between size, yield, and BridgIT weight for berberine. a** A linear pathway comprising four reactions produced benzyl cinnamate. However, the formation of glyoxylate as a byproduct, which could not be used in the subsequent steps, caused a loss of some carbon atoms. **b** While some carbon atoms were lost in linear pathways, branched pathways produced benzyl cinnamate while preserving all carbon atoms and thus had higher yields than linear pathways. **c** Scopolamine could not be produced using linear pathways, as different parts of its complex structure must be derived from different precursors. For example, in this pathway, tropine was derived from putrescine, and tropate was derived from phenylalanine. Then, tropate and tropine were combined to form atropine, which in turn formed scopolamine in two steps. **d** The Pareto fronts for berberine showed that achieving higher yields requires integrating additional reactions. **e** Integrating thermodynamics impacted the Pareto front only for berberine and no other compound (Supplementary Fig. 3). **f** Although the general trend showed that the minimum BridgIT weight increased with size, such an increase was not monotonic; in some cases, lower BridgIT weights were obtained by increasing the size. **g** Integration of thermodynamics indicated the infeasibility of some longer pathways for berberine (for other compounds, see Supplementary Fig. 4). Source data are provided as a Source Data file.

**Table 1 | Comparison of the pathways predicted with SubNetX with the pathways retrieved from the literature**

| Compound | Pathway length | | Predicted molar yield | Pathway # with a better yield than the best-matched pathway |
|---|---|---|---|---|
| | Literature | Predicted | | |
| Benzyl benzoate | 11 | 4–10 | 0.25–0.43 | 81 |
| Benzyl cinnamate | 6 | 4–14 | 0.22–0.38 | 45 |
| *N*-cinnamoyl serotonin | 5 | 5–17 | 0.24–0.32 | 71 |
| Berberine | 13 | 16–26 | 0.13–0.30 | 31 |
| Ajmalicine | 17 | 18–20 | 0.16–0.29 | 66 |
| Quercetin 3-*O*-(6′-acetyl-glucoside) | 9 | 7–12 | 0.21–0.26 | 0 |
| Scopolamine | 20 | 10–28 | 0.19–0.35 | 0 |
| Strictosidine | 15 | 14–22 | 0.14–0.22 | 0 |

sensitivity of our predictions to the choice of weights. We enumerated alternative pathways for each set of weights. Predicting multiple optimal and suboptimal pathways provides experimentalists with a wider range of options, enabling them to explore pathways that may not be immediately apparent. This comprehensive understanding facilitates informed decision-making, allowing researchers to select pathways that may provide better yields, reduce byproducts, or adapt more effectively to varying experimental conditions.

With this weighing system, we first explored the trade-off between the yield and the pathway size by varying the lower bound for the target production to 25%, 50%, 75%, and 100% of the maximum theoretical yield, and for each of these bounds, we computed the minimum pathway size. For all compounds, increasing the product yield beyond a certain threshold increased the pathway size (Fig. 3d and Supplementary Fig. 3). We also investigated how thermodynamic constraints affected the trade-off between yield and size by exploring the Pareto front (Fig. 3e and Supplementary Fig. 3). Here, accounting for thermodynamics did not affect the Pareto optimal front for any compound besides berberine, which had a reduced number of pathways on the Pareto optimal front due to thermodynamic infeasibilities (Fig. 3e). This indicates the inclusion of thermodynamics may alter the trade-off between size and yield, and the shape of the Pareto front. In general, the inclusion of thermodynamics could reduce the optimal yield, suggesting that thermodynamics-based flux analysis[21,22] should always be applied for more reliable predictions.

To further analyze the pathways based on BridgIT weight, we plotted the minimum BridgIT weight of each pathway against pathway size (Fig. 3f and Supplementary Fig. 4). Here, lower BridgIT weights were preferred because they represent a shorter pathway with a higher probability of an appropriate enzyme assignment. The general trend in the rankings showed that the BridgIT weight increased with size, although not monotonically, suggesting that there are some cases where better enzyme assignment can be achieved by increasing pathway size. As before, thermodynamics did not affect the trade-off between size and BridgIT weight, except for berberine (Fig. 3g and Supplementary Fig. 4), as seen in the rankings performed entirely by weight. Also demonstrating the flexibility of this method, the BridgIT weighting system can be modified to further refine the results according to the user's needs by adjusting the coefficients in the objective function. For example, we adjusted the BridgIT weights to give more weight to the relevance of enzyme predictions than to the size of pathways, thus changing the correlation between BridgIT weight and size (Supplementary Fig. 5).

Finally, we evaluated the fraction of thermodynamically feasible pathways for each compound. The highest fraction was found for *N*-cinnamoyl serotonin, where all pathways were feasible. The fraction of thermodynamically feasible pathways was greater than 90% for ajmalicine, benzyl cinnamate, scopolamine, benzyl benzoate, and quercetin 3-*O*-(6′-acetyl-glucoside) (92%, 95%, 97%, 95%, and 99%, respectively). For strictosidine, 90% of the pathways found were feasible, with all infeasibilities occurring on the shortest pathways (Supplementary Fig. 6). However, berberine showed a much lower feasibility, with only 55% of the pathways considered thermodynamically feasible. This is consistent with the observation of a significant change in the Pareto fronts after the integration of thermodynamics.

We further investigated the reason for such a high level of thermodynamic infeasibilities for berberine. We found that all infeasible pathways involved an identical step in which (S)-coclaurine underwent formaldehyde methylation, accompanied by hydrogen peroxide reduction to water. The estimated standard free Gibbs energy for this reaction is highly positive ($22.75 \pm 0.03$ [kcal mol$^{-1}$]). This suggests that by removing the reactions with highly unfavorable standard free Gibbs energies, we can reduce the size of the subnetwork and avoid thermodynamically infeasible pathways. Such considerations are implemented in SubNetX and can be controlled by user-defined parameters.

To assess whether thermodynamic feasibility correlated with the other ranking metrics, we also stratified the pathways based on size or yield and calculated the fraction of feasible pathways in each stratum (Supplementary Fig. 6). The results indicated that for some compounds, the fraction of feasible pathways increased with size (e.g., ajmalicine and berberine), and for some, it increased with yield (e.g., benzyl cinnamate and berberine). However, we could not find a general correlation between thermodynamic feasibility and size or yield that applied to all cases.

## SubNetX predicts pathways for complex non-natural compounds

While the price of chemically synthesized drugs can be prohibitively expensive for patients, biochemical production from simple

**Table 2 | Subnetwork extraction for tadalafil, showing the number of compounds and reactions in the extracted subnetworks for the three settings**

| Metric | Minimal PW | Minimal + 1 | Minimal + 2 |
|---|---|---|---|
| # compounds | 17 | 24 | 2986 |
| # reactions | 21 | 36 | 10,611 |
| # network edges | 16 | 26 | 6198 |
| # isolated network components | 3 | 3 | 1 |
| % compounds in the main component | 53% | 67% | 100% |
| Is the target compound in the main component? | No | No | Yes |

precursors can reduce costs[5,23]. For example, tadalafil is an erectile dysfunction drug produced by chemical synthesis with no known biosynthesis pathways or enzymes in natural organisms[24]. In order for SubNetX to suggest biological alternatives for the synthesis of tadalafil, we first had to assemble the networks, ensuring that all possible reactions for its synthesis were included. Therefore, we first used SubNetX to expand a known pathway previously used for noscapine pathway derivatives[25]. This allows us to expand the biochemical search space so that we do not miss potential alternative pathways with higher yields than the known pathway. We also selected every intermediate in the organic synthesis pathway of tadalafil[26] and extended the network of predicted biochemical reactions to include ones for their synthesis using BNICE.ch, which reconstructs known biochemical reactions as well as generates novel, hypothetical reactions[27]. We added this network to ARBRE to expand the SubNetX search space.

We next used SubNetX to find possible subnetworks for synthesizing tadalafil. To simultaneously determine the impact of adding flexibility to the algorithm in terms of the number of reactions for the synthesis of cosubstrates, we applied SubNetX with three different settings, selecting only the pathways with the (a) minimal number of reaction steps required to connect cosubstrates to the host, (b) minimal + 1 steps, and (c) minimal + 2 steps. To analyze the potential for successful integration of the extracted subnetworks with the host model, we applied topological network analysis (Table 2), and we evaluated whether tadalafil was included in the main component of the network, indicating its connection to the host metabolism. The minimal and minimal + 1-step pathways limited the subnetwork to only 36 reactions and kept tadalafil out of the main component of the subnetwork. Increasing the size of pathways to minimal + 2 extracted a subnetwork of 10,611 reactions with tadalafil integrated into the main component, indicating a potentially successful integration with the host model.

We then evaluated the three extracted subnetworks for yield and feasibility. For the minimal and minimal+1 steps, the model predicted no tadalafil production because the connection between the cosubstrates and the host metabolism was not properly established. In contrast, the minimal + 2 steps subnetwork produced tadalafil with the maximum theoretical yield. We then used this subnetwork to investigate the trade-off between yield and size, as described above for the example syntheses. As with the natural compounds, including more reactions allowed for higher product yields (Fig. 4c). Finding pathways for the synthesis of tadalafil in *E. coli* demonstrates that SubNetX can suggest biosynthesis pathways to produce non-natural compounds for which natural production pathways are not known.

## Discussion

We present SubNetX, a method for exploring large networks of reactions to find an optimal pathway for the synthesis of a target compound. An important feature of our method is the integration of two previously independent approaches to pathway design: graph-based pathway search and constraint-based optimization. The linear pathway search algorithms easily explore large biochemical networks (e.g., ARBRE and ATLASx), and the constraint-based optimization finds the most relevant pathways in terms of yield and feasibility. A second important feature of SubNetX is the extraction of a balanced subnetwork of reactions around the linear pathways from the graph search, which can be seamlessly integrated into the genome-scale metabolic model of any organism. SubNetX takes a broader, more integrative approach to identifying potential pathways than other pathway prediction methods that rely solely on retrobiosynthesis followed by host feasibility filtering[28]. While narrowing the search space early by eliminating infeasible pathways can improve efficiency, it may also limit the discovery of novel or less obvious pathways. SubNetX expands the metabolic network through retrobiosynthesis by connecting cofactors and cosubstrates to the host GEM, then integrates the expanded subnetwork into the host model using MILP. This comprehensive approach allows SubNetX to explore alternative pathways and uncover new opportunities for metabolic engineering, particularly in complex metabolic systems or microbial communities.

Using the SubNetX workflow, we were able to predict pathways for the synthesis of natural products of varying complexity. This work showed that the assembly of multiple reactions into networks from multiple native precursors that converge to product synthesis is necessary for several classes of secondary metabolites, and such pathways could improve production yields and reduce potentially toxic byproducts. We also demonstrated that SubNetX can retrieve experimentally implemented pathways and suggest potentially more efficient alternatives, even in the absence of stoichiometric information or annotation differences. Furthermore, application of SubNetX to a complex non-natural therapeutic, tadalafil, demonstrated its ability to predict biosynthetic pathways for compounds not produced by any known organism.

Experimental investigation and validation of the pathways proposed by SubNetX will allow for its immediate use in a variety of organisms and any application requiring the design and analysis of metabolic pathways. For instance, though our included examples were all in *E. coli*, SubNetX can be readily extended to other host organisms containing a genome-scale model with metabolites annotated using appropriate identifiers. Coverage of gene and metabolite annotations significantly impacts the quality of pathway predictions. As annotation efforts improve, so will the accuracy and scope of genome-scale metabolic models and related prediction methods. Importantly, while GEMs have limitations, the pathways added using SubNetX are guaranteed to be complete, as our method ensures they are elementally balanced, stoichiometrically accurate, and thermodynamically feasible. By integrating SubNetX with highly curated databases such as KEGG and Rhea[29], we aim to mitigate these limitations as much as possible. SubNetX is designed to flexibly incorporate updates in metabolic data, making it well-suited to leverage advancements in genome annotation and metabolic curation. Furthermore, SubNetX is not limited to pathway design for the production of complex molecules and can be used for other applications, such as the biodegradation of xenobiotics[30], designing a synthetic consortium of microorganisms with emergent properties[31,32], or analyzing the interactions between hosts and parasites[33]. Finally, SubNetX is compatible with more advanced constraint-based methods for strain engineering[34–37], allowing the optimization of chassis organisms for industrial applications. We envision that SubNetX will aid in a variety of strain-design optimizations due to its ability to evaluate many possibilities and rank the best solutions based on user-defined criteria.

## Methods

### Selection of the case study compounds

We selected the case studies based on two criteria. First, they should be present in the ARBRE network. Second, they should be labeled as drug-

**Fig. 4 | Pathway prediction for the biosynthesis of tadalafil. a** Tadalafil synthesis pathway used for BNICE.ch network expansion. Atoms of the reactive site are highlighted. Biosynthesis route of piperonal, a precursor of tadalafil (adapted from Jin et al.[43], *enzyme proposed in this work, **enzyme proposed by Jin et al.[43]). 3,4-MDCA: 3,4-methylenedioxycinnamic acid. **b** Chemical synthesis of tadalafil, adapted from Baumann et al.[26] (Step 1) and expanded with necessary intermediates (Steps 2 and 3). The upper path corresponds to the chemical synthesis of Intermediate 2-Cl, and the lower to the proposed biological alternative, Intermediate 2-OH. **c** Exploring the trade-off between size and yield for tadalafil. Source data are provided as a Source Data file.

like compounds based on ChEBI ontology. We randomly chose 70 compounds while preserving the chemical diversity. We also subselected eight compounds which their experimental biosynthesis was available in the literature for further analyses (Supplementary Table 3, Supplementary Data 2–9). The SA score was calculated using the rdkit implementation of the SA score[15].

### Reaction network data
Recently, we generated the largest known network of predicted biochemical reactions, ATLASx[11]. We showed that it is possible to translate reaction networks into graphs connecting at most two compounds per edge and use computational algorithms to navigate these graphs efficiently[38]. In another work, we presented ARBRE, demonstrating that additional data curation and user-defined parameters might refine the pathway ranking to propose only the most relevant pathways as a starting point for experimental tests[13]. ARBRE contains 400,000 reactions, offering a comprehensive yet targeted dataset. This scale represents an optimal balance, allowing for focused and efficient pathway exploration while maintaining the flexibility to incorporate

larger databases, such as ATLASx, when needed. ATLASx and ARBRE integrated the reactions into the reactant-product pair (RP-pair) network. This work used ARBRE as the reaction database to search for the pathways. In the future, the search space for the reactions can be expanded to ATLASx reactions or any other set of balanced reactions.

### Preparing the search space for pathways
We obtained a set of elementally balanced reactions, both known and novel, from ARBRE[13]. For each RP-pair in the network, we attributed all the associated reactions and identified the sets of boundary metabolites. The initial pathway search and the network expansion used the RP-pairs, corresponding reactions, and boundary metabolites. The reactions with their corresponding stoichiometry were stored and integrated into the GEM.

### Translating the reaction network into an explorable hypergraph-like network
The best representation of the structure of biochemical networks is through hypergraphs. However, algorithms to navigate hypergraphs are

not as developed as those to predict linear pathways within a simple graph[2,39]. Simple graphs and linear pathway search algorithms allow assigning weights to graph edges based on atom conservation so that pathways with higher theoretical yields are prioritized[38]. However, while searching for linear pathways, there is no guarantee that the cosubstrates of reactions are also connected to the host metabolism. To preserve the information about all the substrates and products of a reaction within a single edge, as in hypergraphs, we developed a method to integrate cosubstrate information into a simple graph as an edge property. In addition, we introduced a filtering step to exclude unbalanced reactions to avoid problems in pathway evaluation, as unbalanced reactions can produce chemicals without consuming any resources.

### Integrating host metabolites with the reaction network

Host metabolites are regularly annotated in models by public identifiers. We mapped the host metabolites to their corresponding atom connectivity structure without considering cellular compartments. Metabolites without a defined molecular structure were excluded. This allowed us to map the host metabolites to the compounds in ARBRE. We collected the list of metabolites in *E. coli* from iJO1366[18]. Eight hundred of 1099 native metabolites in *E. coli* were mapped to the compounds in ARBRE using the molecular structure. The mapped metabolites then served as precursors for linear pathways.

### Selecting a host organism

SubNetX is applicable to all organisms with available GEMs. To apply SubNetX to other organisms, it is sufficient to map the new host metabolites to the identifiers of the search space of choice, e.g., ARBRE. Here, we applied this pipeline to *E. coli* using the metabolic model iJO1366.

### Controlling the properties of the extracted subnetwork

We have defined a set of parameters that allow users to control the size and coverage of the subnetwork. The user can define the precursor or select from the host native metabolites. Precursor filtering parameters allow for the specification of whether a substructure filtering should be applied. These parameters also define the number of similar precursors to select. A network filtering parameter specifies the threshold for the conserved atom ratio per each pathway step. Users can define the number of linear pathways in the core set and the number of linear pathways to connect cosubstrates. Moreover, there is the option to select only the shortest or the shortest + X-step pathways to connect the cosubstrates, where X is the number of additional steps to include. Users can indicate their preference to include known reactions and longer pathways using the exponential transformation of the graph distances[38].

### Filtering the precursor set based on compound structure

The user-defined set of precursors can be filtered based on chemical substructures. To this end, the user can introduce a *.mol* file describing the desired substructure that should be present in all precursors. We used the rdkit.Chem library (https://www.rdkit.org) to match the substructures. The output of this step is a list of precursors corresponding to the desired structure. The precursors with more carbon atoms than the target (either the primary target or cosubstrates) are removed to achieve an optimal carbon flow. For the remaining precursors, the rdkit.Chem.rdFMCS library is used to calculate the maximum common substructure. The *N* compounds (a user-defined parameter) with the maximum number of atoms in the common substructure are then selected.

### Filtering the network based on user-defined parameters

The network is prepared for the pathway search as described elsewhere[13]. In brief, the unwanted compounds and RP-pairs can be removed from the network based on molecular structures. This network filtering ensures that the pathway search and network expansion

avoid unwanted compounds and use only the reactions with desired properties.

### Finding the core pathway set

The initial linear pathways, subsequently called the core pathway set, consist of linear pathways connecting the target compound to the defined precursors. Such pathways connect the precursors defined in the first step to the target compound. The pathways are found using the k-shortest loopless path algorithm implemented in the networkx library (https://networkx.org). The pathway search algorithm takes user-defined parameters to determine the total number of pathways and the maximum pathway length.

### Subnetwork expansion

Subnetwork expansion is the central step of the pipeline, connecting all the atoms of the target compound to the host metabolism. The initial subnetwork consists of the core pathway set. We associate cosubstrates with the edges of the subnetwork, where each edge represents a step of the pathway. We find linear pathways to connect the non-native metabolites to the host to ensure that all cosubstrates can be produced and all byproducts can be consumed. The new pathways are added to the subnetwork, and new boundary metabolites are specified. The expansion continues until non-native metabolites are connected to the host metabolism. In particular cases, non-native metabolites cannot be connected to the host due to the absence of the pathway in the initial network (i.e., ARBRE) and must be removed in the subnetwork convergence step. We search for reactions with the fewest cosubstrates outside the host metabolism to avoid expanding the subnetwork toward the non-natural cofactor pairs. For each cosubstrate, precursors are selected as described above. If no pathway to the host can be found, the cosubstrate is omitted, and the pathways relying on it will be excluded during the subnetwork convergence. The expansion terminates once all the cosubstrates are either connected to the native metabolites or omitted.

### Subnetwork convergence

We remove the non-native metabolites not connected to the host metabolism during the subnetwork convergence. For this purpose, we remove all the cosubstrates not integrated into the subnetwork and the reactions that require these cosubstrates. Removing these reactions from the subnetwork might disconnect other cosubstrates from the host metabolism. This step is repeated until no cosubstrate out of the subnetwork is left. Thus, we generate a network with a bioproduction or biodegradation pathway for any boundary metabolite. We did not connect metal cofactors, e.g., [4Fe-4S] iron-sulfur cluster, Na⁺, $Co^{2+}$, $Mg^{2+}$, $Cu^{2+}$, and certain gases, e.g., hydrogen selenide, nitrogen, $H_2$, to the host if they were cosubstrates. The reactions and compounds of the converged network are the output that integrates into the host metabolism for further evaluation and ranking.

### Graph visualization

The extracted subnetwork is visualized automatically using the networkx library. Graphs in Gephi format output (.gdf) are generated, which can be imported into Gephi for visualization and further analysis.

### Integration of the subnetwork into the host model

We need to integrate the subnetwork into the host's GEM to find feasible pathways. For this purpose, the metabolites and reactions in the subnetwork should be mapped to their counterparts in the GEM. Then, we construct an integrated network that includes all native and non-native metabolites and reactions. A demand reaction is added for the target to take this compound out of the cytosol, and we set a lower bound for this reaction to ensure that the target is produced. We used the identifiers from our internal database to map the metabolites and reactions. However, other identifiers can be used alternatively if the metabolites and reactions of the GEM are annotated with the same

identifiers. Note that the integration quality depends on the coverage of the annotations, and identifiers with higher coverage are preferred. The integrated subnetwork can be saved in JavaScript Object Notation (JSON) format. JSON is a standard hierarchical format to keep structured data. Since many software packages, such as COBRA[40], support this format, the saved files can be integrated into other pipelines and software for further analyses.

A key feature of SubNetX is its reliance on the first level of InChIKey encoding, which represents the 2D molecular structure of compounds, to unify reaction networks and models. While this approach ensures compatibility across datasets and maintains consistency within the SubNetX framework, it does not account for stereochemistry. However, SubNetX is designed with flexibility to accommodate additional levels of molecular detail. By utilizing full InChIKeys that encode stereochemical data, future applications of SubNetX could incorporate 3D structural information, thereby addressing stereochemical considerations and enhancing the precision of pathway predictions. This capability allows for further refinement of metabolite mapping and reaction analysis, particularly in contexts where stereochemistry plays a critical role in biochemical processes.

## Pathway search and evaluation

The next step in SubNetX is to search for feasible pathways. First, a Flux Balance Analysis (FBA) problem is solved, where the rows and columns of the stoichiometric matrix correspond to the integrated network's metabolites and reactions, and the objective function is to maximize the target production. If the maximum flux of the production is nonzero, at least one feasible pathway exists in the network to produce the target compound. This way, we can also find the maximum production rate of the target ($V_{\text{product}}^{\text{max}}$) for a specified substrate uptake rate. Then, we search for and enumerate the feasible pathways by formulating an MILP problem:

$$\min_{v_j, y_j} \sum y_j$$

$$s.t.: \quad S.v = 0 \tag{1}$$

$$LB_j \le v_j \le UB_j \quad \forall j \in \text{Native} \tag{2}$$

$$y_j LB_j \le v_j \le y_j UB_j \quad \forall j \in \text{Non-native} \tag{3}$$

$$\theta V_{\text{product}}^{\text{max}} \le v_{\text{product}} \tag{4}$$

Here, a binary variable $y_j$ is associated with each non-native reaction. If $y_j$ is zero, the reaction is inactive and not included in the pathway. The objective function is to minimize the number of active non-native reactions. This way, we prioritize feasible pathways with the minimum number of interventions in the host. To ensure that the target compound is produced, Eq. (4) sets a lower bound for the production. This lower bound is set as a fraction ($\theta$) of $V_{\text{product}}^{\text{max}}$ found in the previous step. We can explore the trade-off between yield and the size of the pathway by adjusting $\theta$. While higher values of $\theta$ enforce finding pathways with higher yields, lower values of $\theta$ relax the lower bound and allow us to find shorter pathways.

Integer cut constraints were added to find alternative solutions, including other optimal and suboptimal pathways:

$$\sum_{j \mid y_j^k = 1} \left(1 - y_j\right) \ge 1 \quad \forall k = 1, 2, \ldots, K \tag{5}$$

where $K$ is the total number of pathways found previously, and $y_j^k$ represents if the $k$th pathway includes reaction $j$ or not.

## Ranking feasible pathways

In addition to the saved model as a JSON file, we save a set of tables that contain information about the feasible pathways found by the MILP formulation. These feasible pathways are ranked based on three main criteria. The first criterion is product yield, which indicates how many moles of the target can be produced per consuming one mole of the carbon source, e.g., glucose. Higher yields imply that more carbon is conserved to produce the target.

The second criterion is the weight of the pathway. The weight collectively refers to any type of preference that can be quantified and associated with the reactions. The weight of a pathway is the sum of the scores of its constituting reactions, where pathways with lower weights are preferred. In the simplest case, where all the reactions are weighted equally, minimizing the weight is equivalent to finding pathways with the minimum size. However, we can bias the pathways toward including or not including specific reactions by assigning different weights (e.g., based on toxicity or enzyme specificity).

Finally, the third criterion is thermodynamic feasibility. Some pathways cannot produce the target after integrating thermodynamic constraints due to the changes in reaction directionalities. The output of the thermodynamic evaluation is a binary assignment; the pathways are either thermodynamically feasible or infeasible.

## Pathway weight calculation

Various properties, such as enzyme availability or toxicity of the byproducts, can be quantified as weights and associated with the reactions. Similarly, the weight of a pathway is the sum of the weights of its constituting reactions. In the simplest case, when an equal weight is assigned to all the reactions, minimizing the weight of the pathway is equivalent to minimizing the size. However, we can prioritize including or excluding specific reactions by assigning different weights. For example, we assigned weights to the reactions based on enzyme availability. In particular, we used BridgIT scores, where higher BridgIT scores indicate a higher likelihood of finding an appropriate enzyme. We defined the BridgIT weights as follows:

$$w_j = W - s_j \tag{6}$$

where $s_j$ is the BridgIT score for the $j$th reaction. $W$ is a user-defined parameter, where higher values of $W$ prioritize smaller pathways, and lower values of $W$ prioritize integrating reactions with higher BridgIT scores. We minimized the sum of weights of the active reactions, i.e., $\min_{v_j, y_j} \sum w_j y_j$. A reaction's weight contributes to the objective function only if $y_j = 1$, i.e., the reaction is part of the pathway. We set $W$ to two different values in our analyses, 1.1 and 2, to evaluate the impact of changing $W$.

## Thermodynamic evaluation

We also check the thermodynamic feasibility of the found pathways. To this end, we integrate each pathway into the GEM individually and solve Thermodynamic Flux Analysis (TFA)[21]. Like FBA, TFA optimizes an objective function subject to mass balances. In addition, TFA applies additional constraints so that the directionality of each reaction is determined based on its Gibbs free energy. The Gibbs free energy of a reaction depends on its standard Gibbs free energy, the thermodynamic properties of the environment, and the metabolite concentrations. The latter is approximated based on the available ranges for the metabolite concentrations if metabolomics data is not available[21]. The standard Gibbs free energy of a reaction can be calculated as the difference between the sum of the standard Gibbs free energy of the formation of the products and reactants.

## Thermodynamic curation of the models

Incorporating thermodynamic information allows us to constrain reaction directionality according to the second law of

thermodynamics, specifically by utilizing the Gibbs free energy of each reaction. The thermodynamic curation of the metabolic subnetworks predicted by SubNetX was carried out by calculating the Gibbs free energy of reactions.

We estimated the Gibbs free energy of formation for each compound, along with its associated estimation error. For each metabolite, we retrieved public identifiers (e.g., ChEBI, KEGG, SEED) and used these to obtain compound structures in SMILES format. Using Marvin (version 18.1, 2018, ChemAxon), the SMILES were converted to their primary protonation state at pH 7 and then to MDL Molfiles. These Molfiles were used with the Group Contribution Method to estimate the standard Gibbs free energy of formation for each metabolite and the associated estimation error.

The Gibbs free energy of reactions within the subnetwork was calculated based on the corrected Gibbs free energies of compounds, which were adjusted for pH and ionic strength using the Debye-Hückel approximation[41]. For a reaction with $m$ components, the Gibbs free energy, $\Delta_r G'$, was calculated as:

$$\Delta_r G' = \sum_{j=1}^{m} n_j \Delta_f G_j'^o + RT \ln\left(\prod_{j=1}^{m} x_j^{n_j}\right), \qquad (7)$$

where:

$n_j$: stoichiometric coefficient of compound $j$,

$\Delta_f G_j'^o$ standard Gibbs free energy of formation of compound $j$,

$x_j^{n_j}$: concentration of compound $j$,

$R$: ideal gas constant ($R = 8.31 \cdot 10^{-3} \frac{KJ}{K\,mol}$),

$T$: temperature ($T = 298K$).

In the absence of metabolomics data, generic concentration ranges were used. For intracellular metabolites (i.e., cytoplasmic or periplasmic), concentrations were constrained between 11 and 50 mM, and for extracellular metabolites, between 10 nM and 100 mM[42].

## Calculation of product yield

The product yield was calculated as the number of moles of product generated per mole of substrate (glucose) consumed. To do this, we divided the maximum flux of the exchange reaction to produce this target compound by glucose uptake flux, which was set to 10 mmol h$^{-1}$ gDW$^{-1}$.

## Preparing the database to find pathways for tadalafil

For the first and fourth steps of the proposed biosynthesis pathway of tadalafil (Fig. 4a), we added two new BNICE.ch reaction rules based on EC classes 4.2.1.78 (norcoclaurine synthase) and 4.2.1.145 (capreomycidine synthase), respectively. We identified that the second and third steps could be catalyzed by the reverse reaction of EC 3.4.17.23 (angiotensin-converting enzyme 2) and by 6.3.4.12 (glutamate-methylamine ligase), respectively. One of the precursors of the chemical synthesis of tadalafil, piperonal, is a natural compound, though it does not have a fully characterized biosynthesis route that would connect it to any of the host metabolites in the ARBRE network. To address the biosynthesis of piperonal, we started with the precursor ferulic acid. From this, we added new BNICE.ch rules based on 6.1.3.1 (olefin β-lactone synthetase) to convert ferulic acid into 3,4-methylenedioxycinnamic acid (3,4-MDCA) and the gene PnPNS[43] to convert 3,4-MDCA into piperonal (Fig. 4b). In all, expanding the BNICE.ch network and identifying all the reactions carrying flux from tryptophan, as the native precursor, toward tadalafil added eight auxiliary reactions to the ARBRE network. This ensured that all the necessary reactions were present to reconstruct a balanced subnetwork to synthesize tadalafil.

## Reporting summary

Further information on research design is available in the Nature Portfolio Reporting Summary linked to this article.

## Data availability

The extracted subnetworks and models for all compounds are available in Zenodo[44] [https://doi.org/10.5281/zenodo.15119707]. The list of the 70 compounds used in this study is provided as Supplementary Data 1. Supplementary Data 2–9 present the comparison between predicted pathways in this work and the experimental implementation for various compounds. Source data are provided with this paper.

## Code availability

Subnetwork extraction and analysis code are available at Github [https://github.com/EPFL-LCSB/SubNetX]. The code is also deposited in Zenodo to provide a reference to the version used for this study[45]. The SubNetX code was tested on macOS and Linux systems. For this study, we used Mac Pro machines with 32 GB of RAM and a 3.7 GHz Quad-Core processor. Installation requires Python 3.7 with the following packages: networkx, rdkit, cobra, and pytfa. The project's GitHub repository provides a detailed installation guide and all dependencies. The optimization problems were solved using the commercial solver ILOG CPLEX, version 12.8.0.

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

## Acknowledgements
We would like to thank Dr Kaycie Butler and Dr Ljubisa Miskovic for their valuable comments on the language and structure of the manuscript. This work was funded by the Swiss National Science Foundation (SNSF) under grant 200021_188623 (A.S. and O.O.), by the European Union's Horizon 2020 Research and Innovation Program under grant agreement no. 814408 (A.S. and O.O.), by the NCCR Microbiomes, a National Centre of Competence in Research under grant number 180575 (O.O.), and by the École Polytechnique Fédérale de Lausanne.

## Author contributions
A.S., O.O. and V.H. designed and conceptualized the study and developed the methodology. A.S. implemented the software and ran the simulations to extract subnetworks. O.O. implemented the software and ran the simulations to evaluate and rank the pathways. A.S. validated the predicted pathways with the reported pathways in the literature. A.S. and O.O. curated the data, developed the visualizations, performed formal analysis, conducted the investigation, and wrote the original draft. A.S., O.O., and V.H. discussed and analyzed the results. A.S., O.O. and V.H. wrote and reviewed the manuscript. V.H. was responsible for acquiring resources, funding, project administration, and supervision.

## Competing interests
The authors declare no competing interests.
