## [Peer Review file · Nature Communications]

Designing pathways for bioproducing complex chemicals by combining tools for pathway extraction and ranking

Corresponding Author: Professor Vassily Hatzimanikatis

Version 0:

Reviewer comments:

Reviewer #1

(Remarks to the Author)

Sveshnikova et al proposed SubNetX to design pathways for complex chemicals, with constraints of the metabolic network, cofactors and host organism. It is of significance to the biosynthesis design since the host organism is not considered in most of the retro-biosynthesis approaches. And several cases were investigated to show the ability to predict biosynthetic pathways for different types of compounds. However, the contributions of this work do not seem to be outstanding. The reaction network (ARBRE or ATLASx) and genome-scale metabolic models have been published elsewhere. The pathway search and expansion are also not something new either. The innovation in algorithms should be elaborated explicitly. And this work also lacks a comprehensive evaluation and discussion to demonstrate the accuracy of the model in predicting biosynthetic pathways.

Here are the major concerns:

1. Although SubNetX makes predictions based on the predefined reaction network, considering that a considerable portion of the network consists of predicted reactions, the recent bio-retrosynthesis planning approaches (for example, see Nat Catal 6, 137–151 (2023). <https://doi.org/10.1038/s41929-022-00909-w>) should be mentioned and compared. The authors did not even compare with any known methods in the current manuscript.
2. The authors discussed a total of nine cases in this work, larger scale testing is necessary to support the conclusions presented in this work, for example L145, L170 and L178. In this work, SubNetX mainly based on ARBRE (designed for aromatic compounds), the generalizability of this approach is concerned. More extensive evaluation is needed, unless SubNetX is described as aromatics-specific.
3. Table 1, none of the reported pathways is recovered, the reason should be addressed. If experimental pathways were implemented not in *E. coli*, why were they selected for testing? If the experimentally implemented pathways might include reactions with unknown mechanisms or stoichiometries, this might be the shortage of SubNetX, then more discussion is required.
4. The results in this work are confusing, the meaning of many graphs, such as Fig. 2b&c, Fig. d-g and Fig. 4c seems not clear, what is their relationship with the ability of predict biosynthetic pathways? More analysis is required to demonstrate the pathway prediction ability of SubNetX, especially the accuracy to recover experimentally validated pathways, the feasibility or diversity.

Minor comments:

1. The manuscript lacks necessary background and method description, for example, the introduction of thermodynamic GEM and the calculation of yield.
2. Supplementary Figure 1: it is difficult for me to read. More explanatory text would greatly aid in understanding the network extraction.
3. Is stereochemistry addressed in ARBRE or ATLASx, since it is essential to the application of predicted pathways.
4. As much as I know, the GEMs are not so complete, even for *E. coli*, many metabolites and reactions are still missing. This clearly limits the prediction ability of SubNetX, which should be discussed in the manuscript.

(Remarks on code availability)

Reviewer #2

(Remarks to the Author)

Here the authors present a new computational tool called SubNetX that combines the strength of constraint-based and retrobiosynthesis methods to design new pathways for bioproduct biosynthesis. SubNetX uses constraint-based methods to evaluate yield and ensure thermodynamic feasibility, while using retrobiosynthesis to propose novel pathways.

This is fantastic work addressing critical challenges associated with the combination of genome-scale metabolic models and cheminformatics-generated hypothetical reaction databases to predict novel reactions. It is particularly timely as well, because this paper is coming at a time when breakthroughs in enzyme structure prediction, robotics, metabolomics, proteomics, and AI are making it possible to actually engineer these pathways. One of the great challenges in this field is the computational challenge associated with the powerful stoichiometric constraint-based approaches, generally making these approaches intractable to the massive scale of cheminformatics databases. The authors resolve this problem with their innovative network expansion approach, which subselects the relevant portion of the source database reactions for inclusion in the selected host model. This makes the powerful constraint-based approaches tractable. AI approaches may emerge that achieve similar results, but likely only at the cost of vastly more compute (particularly GPUs), rendering such approaches to be of limited accessibility to most potential use.

The manuscript is well-written for the most part. I have only a few comments or concerns outlined below:

1.) The authors report on page 7: "We found that for all eight compounds there was at least one feasible pathway to produce them with the maximum theoretical yield of 100% g-C, without loss of carbon to other byproducts." This is surprising. Are these truly 100% yield pathways? Meaning they involve no redox cost or ATP cost? If there are currency metabolite costs, it might be good to mention these.

2.) In the alternative pathways described on page seven, the yields were higher due to the avoided production of useless byproducts, but again, did this result in additional cost in currency cofactors (e.g. ATP, NADH)?

3.) It's not entirely clear from the manuscript, but I am assuming here the authors are proposing to use predicted novel reactions from cheminformatics pipelines. Is this true? If so, it might be good to clarify this in the text, as this does create new challenges in any attempt to engineer these pathways. That said, with modern AI, structural analysis, alphafold etc, this is far less of a challenge today than it has been previously.

4.) On page 8, the authors state "had 6 out of 12 reactions in common with the experimentally implemented pathway, matching the first three levels of the EC numbers". This language seems confusing. 6 of the 12 "reactions" are not identical, correct? They do not have identical reactants and products? They have identical reaction classes? Shouldn't the sentence then say, "6 out of 12 reactions classes in common". The manuscript does this a lot - interchangeably talking about novel predicted reactions and experimentally determined reactions using common terms.

5.) Although there is a citation, it would be good to have a very short description following the first mention of ARBRE and ATLASx in the results, as this clarifies the database behind the SubNetX algorithm. It might also even be useful to mention how big ARBRE and ATLASx are so the reader appreciates the scale of data that the author's method is actually coping with. It may seem like a distraction, but this speaks to the power and significance of SubNetX.

6.) How were thermodynamic properties computed? What concentration ranges were considered in thermodynamic constraints? Were experimental concentration values used?

7.) In mapping metabolites in ARBRE and ATLASx to host models in the methods, structure was used. Was stereochemistry considered? Generally, many cheminformatics-generated reactions are blind to stereochemistry, making union with metabolic models a challenge.

8.) What kind of linear optimization solvers were used for the work? I would suspect open-source solvers would still not be up to solving many portions of the SubNetX workflow, but it would be good to report this.

(Remarks on code availability)

The code is in a well-documented public git repository. There are example parameter files for all bioproducts studied in the paper. There is also a default file for users to try their own compounds. There are clear installation instructions as well, and the dependencies are very reasonable.

Reviewer #3

(Remarks to the Author)

The authors present a new method for combining the strengths of retrobiosynthesis and constraint-based pathway design. The paper is well written, and presents a useful method that would be broadly useful to the field. The work does support the conclusion, and open source software is provided to reproduce these methods. I thank the authors for putting in the effort to provide well documented code to allow others to leverage this method.

I have a few questions and suggestions for the authors:

1) What external data sources or software tools were required to do the full analysis presented here, and are any of them proprietary or commercial? For example, it mentions that BNICE.ch was integrated to include novel hypothetical reactions, but the code repository says the data source is ARBRE. It would be helpful to provide some information on what is required

to use the code, such as system requirements or other software and data.

2) To what extent does this differ methodologically, and compare performance wise to the previously published GEM-Path method, which integrates retrobiosynthesis with genome scale models (PubMed ID 25080239)?

3) Although it is impressive how well this works for the 8 compounds listed, it would be extremely valuable to perform this analysis on a large set of molecules, such as all FDA approved compounds, and get detailed aggregate statistics. 8 compounds is a small set and may not be very representative. If possible, a quantitative comparison to some existing widely used retrobiosynthesis tools would be valuable.

(Remarks on code availability)

The code is freely available, and appears well documented, and well written. I did not attempt to install and run the code to test it myself.

Version 1:

Reviewer comments:

Reviewer #1

(Remarks to the Author)

The authors addressed some of my issues, while there are still two major concerns need to be addressed:

1. I understand that large-scale testing consumes time and computational resources, but I still believe that additional case studies such as synthetic drugs and other categories of natural products are necessary.
2. None of the experimental pathways is recovered, this should be further discussed. Do all experimental pathways contain unbalanced reactions? In an ideal situation, SubNetX could predict pathways with reactions from any organism, including E. coli, so it "should" predict the reported pathways. By the way, there is no Supplementary Table 4 in the Supplementary Information.

(Remarks on code availability)

Reviewer #2

(Remarks to the Author)

The authors have responded rigorously to all of my prior comments, making many clarifying changes to the manuscript. As such, I have no new comments or suggestions at this time.

(Remarks on code availability)

No changes from prior comments.

Reviewer #3

(Remarks to the Author)

While the authors addressed most of my concerns, I still feel that 8 compounds is not a large enough set to systematically evaluate the ability of the method to generalize to diverse targets. The authors mention using an older computer system which can process a single compound between 40 minutes and 8 hours, which would allow for a test set in one week of between 21 and 252 compounds. My suggestion would be to run at least as many as can be run in a full week, with an attempt to maximize the chemical diversity of the test set in an unbiased way.

The authors have thoroughly addressed all of my other questions and concerns.

(Remarks on code availability)

The code is freely available, and appears well documented, and well written. I did not attempt to install and run the code to test it myself.

Version 2:

Reviewer comments:

Reviewer #1

(Remarks to the Author)

well addressed

(Remarks on code availability)

Reviewer #3

(Remarks to the Author)

The authors have addressed all of my issues, and I thank them for expanding their analysis to a greater number of compounds.

However, I would like to point out that the new supplemental file is not fully readable: the columns are too narrow and the full chemical names and chemical formulas cannot be read. I request that the authors upload a fixed version of this file, but otherwise I think the manuscript is in good shape, and ready to publish.

(Remarks on code availability)

I have no new comments on the code since previous review.

Authors response to the editor and reviewers' comments

We thank the reviewers and the editor for their insightful, constructive, and positive comments. We have highlighted below our detailed responses to the suggested improvements, suggestions, and changes.

Reviewer #1 (Remarks to the Author):

Sveshnikova et al proposed SubNetX to design pathways for complex chemicals, with constraints of the metabolic network, cofactors and host organism. It is of significance to the biosynthesis design since the host organism is not considered in most of the retro-biosynthesis approaches. And several cases were investigated to show the ability to predict biosynthetic pathways for different types of compounds. However, the contributions of this work do not seem to be outstanding. The reaction network (ARBRE or ATLASx) and genome-scale metabolic models have been published elsewhere. The pathway search and expansion are also not something new either. The innovation in algorithms should be elaborated explicitly. And this work also lacks a comprehensive evaluation and discussion to demonstrate the accuracy of the model in predicting biosynthetic pathways.

We sincerely thank the reviewer for acknowledging the importance of our work and its potential significance in biosynthesis design. While we understand the reviewer's perspective regarding the novelty of certain individual components of our approach, we would like to emphasize that the primary innovation lies in the integration and application of these components to create a comprehensive, biochemically informed framework.

Our approach uniquely incorporates mechanistic and thermodynamic considerations, which are often overlooked in retrobiosynthesis methodologies. By constructing a hypergraph-like network as an intermediate step, we define a hypothetical feasible space that connects target molecules to the host organism's native metabolism while embedding mechanistic details such as reaction thermodynamics and kinetics. This novel framework is designed not only for pathway design but also as a physics-informed component that can enhance AI-based retro-biosynthesis pipelines by introducing biochemically grounded insights.

To address the reviewer's concern, we have expanded the Introduction to explicitly highlight this contribution, previously detailed in the Methods section. The added paragraph reads as follows:

"The innovation of our algorithm lies in assembling a hypergraph-like network as an intermediate step in pathway design. This network defines a feasible solution space that connects a target molecule to the native metabolism of the host organism. Crucially, it integrates mechanistic details, including thermodynamics and kinetics, to enhance the reliability and precision of pathway predictions."

We also refer the reviewer to the response we provided to the second comment by the third reviewer.

Additionally, we have tried to address the reviewer's concerns regarding the evaluation and discussion of the model's accuracy in predicting biosynthetic pathways, as detailed in subsequent responses.

Here are the major concerns:

1. Although SubNetX makes predictions based on the predefined reaction network, considering that a considerable portion of the network consists of predicted reactions, the recent bio-retrosynthesis planning approaches (for example, see *Nat Catal* 6, 137–151 (2023). <https://doi.org/10.1038/s41929-022-00909-w>) should be mentioned and compared. The authors did not even compare with any known methods in the current manuscript.

We thank the reviewer for highlighting the relevance of recent retrobiosynthesis approaches, including the one referenced. The retrobiosynthesis tool employed in this study is BNICE.ch, which shares similarities with some of the methods mentioned. However, BNICE.ch has already been published, and its comparison with other retrobiosynthesis tools is beyond the scope of this study. Notably, the core contribution of SubNetX is independent of the reaction prediction tool used, as SubNetX is agnostic to the source of reaction data. It functions effectively based on the scope and quality of the provided network, regardless of whether the network includes predicted or experimentally verified reactions.

To our knowledge, no existing method explicitly assembles hypergraph-like networks to mechanistically explore the connections between target compounds and host metabolism nor integrates these networks with detailed thermodynamic assessments to evaluate pathway feasibility. SubNetX represents a novel framework in this regard.

In this study, we chose to benchmark SubNetX using ATLASx, as it is the largest published network of biochemical reactions and was recently presented in *Nature Communications*. This choice also highlights the compatibility of our framework with large-scale networks while illustrating its flexibility to operate with smaller databases, albeit with limitations in the scope of predictions proportional to database size.

To clarify this, we have added the following sentence to the Introduction:

"To the best of our knowledge, no other approach attempts to mechanistically explore the potential connections of a target compound to the host metabolism by assembling such networks and then using this network as an input to integrate mechanistic details such as thermodynamics to assess feasibility."

We hope this additional context addresses the reviewer's concerns.

2. The authors discussed a total of nine cases in this work, larger scale testing is necessary to support the conclusions presented in this work, for example L145, L170 and L178. In this work, SubNetX mainly based on ARBRE (designed for aromatic compounds), the generalizability of this approach is concerned. More extensive evaluation is needed, unless SubNetX is described as aromatics-specific.

SubNetX is a search algorithm and workflow that is independent of the input database of biochemical reactions. We used ARBRE to reconstruct the network, while we have also specifically expanded the reaction network using BNICE.ch outside of the scope of ARBRE

for several compounds presented in this paper (scopolamine and tadalafil). We selected the ARBRE database for this study because it provides a highly curated and stoichiometrically balanced collection of reactions, ensuring reliable integration with SubNetX for pathway prediction. ARBRE's focus on aromatic compounds aligns with our goal of designing biosynthetic pathways for industrially relevant products, offering a comprehensive yet targeted dataset of approximately 400,000 reactions. This scale represents an optimal balance, allowing for focused and efficient pathway exploration while maintaining the flexibility to incorporate larger databases, such as ATLASx, when needed. Additionally, ARBRE includes both experimentally validated and cheminformatics-predicted reactions, enabling SubNetX to explore novel pathways while ensuring accuracy. Its compatibility with thermodynamic and stoichiometric constraints makes it ideal for generating meaningful and innovative insights in biosynthetic design. Furthermore, ARBRE is only 2% (8'000 reactions) biased towards aromatic compounds while including all the basic metabolic pathways.

We added the following phrase to the Methods:

"ARBRE contains 400,000 reactions, offering a comprehensive yet targeted dataset. This scale represents an optimal balance, allowing for focused and efficient pathway exploration while maintaining the flexibility to incorporate larger databases, such as ATLASx, when needed."

3. Table 1, none of the reported pathways is recovered, the reason should be addressed. If experimental pathways were implemented not in *E. coli*, why were they selected for testing? If the experimentally implemented pathways might include reactions with unknown mechanisms or stoichiometries, this might be the shortage of SubNetX, then more discussion is required.

We appreciate the reviewer's feedback and hope this response addresses their concerns:

Our study used *E. coli* as the primary model organism to maintain consistency and evaluate the recovery of heterologous pathways described in the literature despite *E. coli*'s inherent limitations in specific steps.

The ARBRE database used in this study includes only elementally balanced reactions with defined stoichiometries. In contrast, many reactions from other databases or publications lack this level of curation, posing challenges for pathway recovery. While addressing these issues is beyond the scope of this work, we recognize their impact on pathway modeling while emphasizing the following notes:

1. Even with *E. coli* as the host organism, SubNetX predicted pathways with reactions from other organisms. Properly curated, balanced reactions from these pathways can be included in the model.
2. Missing reactions often fail to meet the criteria of being elementally balanced or represent generalized class reactions instead of specific biochemical transformations.

We acknowledge that unbalanced reactions present challenges even in experimentally validated pathways. SubNetX supports highly curated databases, such as KEGG or Rhea, to incorporate pathways into high-quality metabolic models. We manually reviewed the pathways and documented specific issues in Supplementary Table 4, providing a detailed discussion of the findings. Additionally, we acknowledge the need for more analysis to demonstrate the ability of SubNetX to recover experimentally validated pathways, assess pathway feasibility, and evaluate pathway diversity. We address these aspects in more detail in Supplementary Table 4, providing a comprehensive evaluation of pathway recovery and discussing specific cases where experimental pathways were successfully recovered or identified limitations.

4. The results in this work are confusing, the meaning of many graphs, such as Fig. 2b&c, Fig. d-g and Fig. 4c seems not clear, what is their relationship with the ability of predict biosynthetic pathways? More analysis is required to demonstrate the pathway prediction ability of SubNetX,

especially the accuracy to recover experimentally validated pathways, the feasibility or diversity.

Figure 2b shows the dynamics of subnetwork expansion. The number of boundaries represents the biochemical scope the algorithm explores to construct candidate biosynthetic pathways. Peaks in the number of boundaries indicate the addition of intermediates or side branches, which are critical for evaluating alternative pathways and ensuring coverage of possible reactions. Figure 2c illustrates how the iterative refinement process narrows the extended network to identify plausible biosynthetic pathways. The rapid reduction in boundaries means that redundancies or unlikely pathways are removed, resulting in a focused and predictive set of reactions. By presenting these metrics, Figures 2b and 2c quantitatively demonstrate the effectiveness of the network extraction and convergence in building accurate biosynthetic pathway predictions. To better reflect this in the paper, we edited the Figure 2 caption:

a The chemical structures of each studied compound are shown alongside their synthetic accessibility scores, which evaluate how easily these compounds can be synthesized. **b** Network extraction: The number of boundaries (representing potential intermediate or limiting compounds) identified during each round of subnetwork extraction is plotted. This metric reflects the extent to which the reaction network expands, capturing the relevant metabolic context for predicting biosynthetic pathways. Peaks in this graph highlight the addition of important intermediates or branches to the network. **c** Network convergence: The progressive reduction in the number of boundaries across successive rounds of refinement is shown. This demonstrates how the algorithm systematically narrows the extended network by removing redundant or unlikely pathways, ultimately focusing on plausible biosynthetic routes.

We are also pleased to clarify the purpose of the analysis presented in Fig. 3d-g and Fig. 4c. In these figures, we investigated the sensitivity of our approach to variations in input parameters, specifically the weights assigned to different objective functions. This analysis allowed us to explore the tradeoffs between engineering objectives, such as achieving higher yields and minimizing pathway length. Additionally, we demonstrated the ability of our framework to enumerate both optimal and suboptimal pathways. Identifying multiple pathways, including suboptimal ones, gives experimentalists a broader range of options. This flexibility enables them to consider metabolic pathways that may not be immediately apparent but could offer advantages in specific experimental contexts. Such insights empower researchers to make more informed decisions when selecting pathways, optimizing for higher yields, minimizing byproducts, or adapting to varying experimental conditions.

To reflect this analysis in the manuscript, we have added the following sentences:

"Our goal was to investigate the tradeoffs between different engineering objectives, such as higher yield and shorter pathway length, while analyzing the sensitivity of our predictions to the choice of weights. We enumerated alternative pathways for each set of weights. Predicting multiple optimal and suboptimal pathways provides experimentalists with a wider range of options, enabling them to explore pathways that may not be immediately apparent. This comprehensive understanding facilitates informed decision-making, allowing researchers to select pathways that may provide better yields, reduce byproducts, or adapt more effectively to varying experimental conditions."

We hope this addition addresses the reviewer's concerns and provides the necessary clarity.

Minor comments:

1. The manuscript lacks necessary background and method description, for example, the introduction of thermodynamic GEM and the calculation of yield.

We thank the reviewer for pointing out the need for additional background and methodological details. To address this, we have added the following explanation to the manuscript regarding the thermodynamic constraints and yield calculation:

"Thermodynamic Curation of the Models

Incorporating thermodynamic information allows us to constrain reaction directionality according to the second law of thermodynamics, specifically by utilizing the Gibbs free energy of each reaction. The thermodynamic curation of the metabolic subnetworks predicted by SubNetX was carried out in two steps:

1. Gibbs Free Energy of Formation

We estimated the Gibbs free energy of formation for each compound, along with its associated estimation error. For each metabolite, we retrieved public identifiers (e.g., ChEBI, KEGG, SEED) and used these to obtain compound structures in SMILES format. Using Marvin (version 18.1, 2018, ChemAxon), the SMILES were converted to their primary protonation state at pH 7 and then to MDL Molfiles. These Molfiles were used with the Group Contribution Method (GCM) to estimate the standard Gibbs free energy of formation for each metabolite and the associated estimation error.

2. Gibbs Free Energy of Reactions

The Gibbs free energy of reactions within the subnetwork was calculated based on the corrected Gibbs free energies of compounds, which were adjusted for pH and ionic strength using the Debye-Hückel approximation. For a reaction with m components, the Gibbs free energy, $\Delta_r G'$, was calculated as:

$$\Delta_r G' = \sum_{j=1}^m n_j \Delta_f G_j'^o + RT \ln \left(\prod_{j=1}^m x_j^{n_j} \right), \quad (7)$$

where:

- n_j : stoichiometric coefficient of compound j ,
- $\Delta_f G_j'^o$: standard Gibbs free energy of formation of compound j ,
- $x_j^{n_j}$: concentration of compound j ,
- R : ideal gas constant ($R = 8.31 \cdot 10^{-3} \frac{KJ}{K mol}$),
- T : temperature ($T = 298 K$).

In the absence of metabolomics data, generic concentration ranges were used. For intracellular metabolites (i.e., cytoplasmic or periplasmic), concentrations were constrained between 11 μ M and 50 mM, and for extracellular metabolites, between 10 nM and 100 mM.

Calculation of Product Yield

The product yield was calculated as the number of moles of product generated per mole of substrate (glucose) consumed. To do this, we divided the maximum flux of the exchange reaction to produce this target compound by glucose uptake flux, which was set to 10 mmol h⁻¹ gDW⁻¹.

We hope this addition provides sufficient background and clarification on the thermodynamic model and yield calculation methodology.

2. Supplementary Figure 1: it is difficult for me to read. More explanatory text would greatly aid in understanding the network extraction.

We thank the reviewer for pointing out the difficulty in interpreting Supplementary Figure 1. To address this, we have simplified the figure by retaining only the essential information and reorganizing its layout to align more closely with the figure description in the text. Additionally, we have ensured that the Methods section now includes a more detailed explanation of the network extraction process to complement the visual information in the figure. This explanation provides step-by-step details to clarify how the network is constructed and refined during pathway prediction. We hope these improvements make the figure and its associated content easier to understand and more accessible to the reader.

3. Is stereochemistry addressed in ARBRE or ATLASx, since it is essential to the application of predicted pathways.

We thank the reviewer for raising the important issue of stereochemistry. As described in the ATLASx paper, which also applies to ARBRE, we added the following text:

"A key feature of SubNetX is its reliance on the first level of InChIKey encoding, which represents the 2D molecular structure of compounds, to unify reaction networks and models. While this approach ensures compatibility across datasets and maintains consistency within the SubNetX framework, it does not account for stereochemistry. However, SubNetX is designed with flexibility to accommodate additional levels of molecular detail. By utilizing full InChIKeys that encode stereochemical data, future applications of SubNetX could incorporate 3D structural information, thereby addressing stereochemical considerations and enhancing the precision of pathway predictions. This capability allows further refinement of metabolite mapping and reaction analysis, particularly in contexts where stereochemistry plays a critical role in biochemical processes."

4. As much as I know, the GEMs are not so complete, even for *E. coli*, many metabolites and reactions are still missing. This clearly limits the prediction ability of SubNetX, which should be discussed in the manuscript.

We appreciate the reviewer's comment and agree that genome-scale metabolic models (GEMs) are not fully complete, even for well-studied organisms like *E. coli*. This incompleteness indeed poses challenges for pathway prediction methods, including SubNetX. However, it is important to emphasize that this limitation is a universal challenge in systems biology, not one specific to GEMs or our approach. The incompleteness of GEMs is primarily due to gaps in gene annotations, which directly affect the identification and representation of metabolic reactions and pathways. Many metabolites and reactions remain undocumented, not because the models themselves are inherently flawed but because the underlying biological knowledge is still evolving. Improvements in gene annotation, metabolite characterization, and reaction documentation will enhance the predictive capabilities of all computational tools that rely on these datasets, including SubNetX.

To address this point in the manuscript, we have added the following statement:

"Coverage of gene and metabolite annotations significantly impacts the quality of

pathway predictions. As annotation efforts improve, so will the accuracy and scope of genome-scale metabolic models and related prediction methods. Importantly, while GEMs have limitations, the pathways added using SubNetX are guaranteed to be complete, as our method ensures they are elementally balanced, stoichiometrically accurate, and thermodynamically feasible. By integrating SubNetX with highly curated databases such as KEGG and Rhea, we aim to mitigate these limitations as much as possible. SubNetX is designed to flexibly incorporate updates in metabolic data, making it well-suited to leverage advancements in genome annotation and metabolic curation.”

We hope this explanation clarifies the scope of the issue and places it in the broader context of ongoing improvements in systems biology.

Reviewer #2 (Remarks to the Author):

Here the authors present a new computational tool called SubNetX that combines the strength of constraint-based and retrobiosynthesis methods to design new pathways for bioproduct biosynthesis. SubNetX uses constraint-based methods to evaluate yield and ensure thermodynamic feasibility, while using retrobiosynthesis to propose novel pathways.

This is fantastic work addressing critical challenges associated with the combination of genome-scale metabolic models and cheminformatics-generated hypothetical reaction databases to predict novel reactions. It is particularly timely as well, because this paper is coming at a time when breakthroughs in enzyme structure prediction, robotics, metabolomics, proteomics, and AI are making it possible to actually engineer these pathways. One of the great challenges in this field is the computational challenge associated with the powerful stoichiometric constraint-based approaches, generally making these approaches intractable to the massive scale of cheminformatics databases. The authors resolve this problem with their innovative network expansion approach, which subselects the relevant portion of the source database reactions for inclusion in the selected host model. This makes the powerful constraint-based approaches tractable. AI approaches may emerge that achieve similar results, but likely only at the cost of vastly more compute (particularly GPUs), rendering such approaches to be of limited accessibility to most potential use.

The manuscript is well-written for the most part. I have only a few comments or concerns outlined below:

1.) The authors report on page 7: "We found that for all eight compounds there was at least one feasible pathway to produce them with the maximum theoretical yield of 100% g-C, without loss of carbon to other byproducts."

This is surprising. Are these truly 100% yield pathways? Meaning they involve no redox cost or ATP cost? If there are currency metabolite costs, it might be good to mention these. We thank the reviewer for their insightful observation and the opportunity to clarify this point. In our simulations, we assumed that glucose served as the limiting substrate, consistent with its widespread use as the primary carbon source in industrial settings. The uptake of inorganics, including ions, was assumed to be unrestricted, reflecting conditions where such compounds are available in excess. However, apart from the uptake of ammonia required to supply nitrogen for the biosynthesis of nitrogen-containing compounds (e.g., ajmalicine, N-cinnamoyl serotonin, berberine, scopolamine, and strictosidine), no uptake of other inorganic compounds was observed.

This suggests that the redox cost for synthesizing ATP, NADH, or NADPH, as well as any other chemical energy requirements, was met entirely through glucose metabolism. The absence of additional inorganic uptake implies that all necessary cofactors and reducing equivalents were generated within the constraints of glucose as the sole carbon source.

To address this in the manuscript, we added the following clarification:

"As in previous studies, we assumed glucose to be the limiting substrate by constraining its uptake (see Methods), while inorganic compounds were assumed to be available in excess. Notably, aside from the uptake of ammonia to provide nitrogen for the synthesis of nitrogen-containing compounds (ajmalicine, N-cinnamoyl serotonin, berberine, scopolamine, and strictosidine), no uptake of other inorganics was observed in different alternative optimal pathways. This finding indicates that glucose uptake alone was sufficient to meet the chemical and redox costs required for target compound synthesis, including the production of ATP, NADH, and NADPH."

2.) In the alternative pathways described on page seven, the yields were higher due to the avoided production of useless byproducts, but again, did this result in additional cost in currency cofactors (e.g. ATP, NADH)?

The explanation provided in the previous comment applies to the alternative pathways too. In our analysis of alternative pathways, the higher yields observed were attributed to avoiding byproduct formation, allowing more carbon from the substrate to be directed to the target compound. Importantly, these higher yields were achieved without incurring additional costs in currency cofactors such as ATP, NADH, or NADPH.

We clarified this in the text through the sentences we added for the previous comment.

3.) It's not entirely clear from the manuscript, but I am assuming here the authors are proposing to use predicted novel reactions from cheminformatics pipelines. Is this true? If so, it might be good to clarify this in the text, as this does create new challenges in any attempt to engineer these pathways. That said, with modern AI, structural analysis, alphafold etc, this is far less of a challenge today than it has been previously.

We thank the reviewer for pointing out the importance of clarifying this aspect of our work. Our pipeline incorporates known and predicted reactions derived from cheminformatics-based retrosynthesis approaches. These predicted reactions significantly expand the biochemical search space, enabling SubNetX to propose hypothetical biosynthetic pathways that may not yet have been experimentally validated or observed. This approach introduces specific challenges, such as the need to validate enzyme activity, reaction feasibility, and compatibility with host metabolism. However, advances in modern tools and technologies, including AI-driven structural modeling (e.g., AlphaFold), enzyme engineering, and thermodynamic modeling, mitigate many of these difficulties. These tools provide a framework for assessing enzyme specificity and activity, increasing confidence in the viability of predicted pathways and reducing the experimental effort required for validation.

To ensure clarity, we have added the following to the manuscript discussion:

"The inclusion of cheminformatics-predicted reactions allows SubNetX to identify novel pathways with potentially higher yields than those reported experimentally. By expanding the biochemical search space, these pathways offer innovative solutions for biosynthetic design while highlighting the necessity of experimental validation to address uncertainties in enzyme specificity and reaction mechanisms. Advances in structural modeling and validation tools, such as AlphaFold, can enhance the reliability of these predictions by assessing enzyme compatibility and reaction feasibility. This integration of cheminformatics and machine learning tools exemplifies the increasing feasibility of exploring and engineering hypothetical pathways in practical applications."

4.) On page 8, the authors state "had 6 out of 12 reactions in common with the experimentally implemented pathway, matching the first three levels of the EC numbers". This language seems confusing. 6 of the 12 "reactions" are not identical, correct? They do not have identical reactants and products? They have identical reaction classes? Shouldn't the sentence then say, "6 out of 12 reactions classes in common". The manuscript does this a lot - interchangeably talking about novel predicted reactions and experimentally determined reactions using common terms.

We thank the reviewer for highlighting this issue and pointing out the potential confusion in the terminology. You are correct that the term "reactions" may imply identical reactants and products, which is not the case here. Instead, we were referring to reaction classes, as defined

by the first three levels of the Enzyme Commission (EC) numbers, which classify reactions based on their catalytic mechanism rather than specific substrates or products.

To address this, we have revised the sentence for clarity as follows:

"6 out of 12 reaction classes were shared with the experimentally implemented pathway, based on matching the first three levels of the EC numbers."

We have also reviewed the manuscript to ensure consistent terminology, distinguishing between individual reactions (with specific reactants and products) and reaction classes (defined by EC numbers). This ensures that readers can differentiate between experimentally observed reactions and predicted reaction classes, avoiding confusion.

5.) Although there is a citation, it would be good to have a very short description following the first mention of ARBRE and ATLASx in the results, as this clarifies the database behind the SubNetX algorithm. It might also even be useful to mention how big ARBRE and ATLASx are so the reader appreciates the scale of data that the author's method is actually coping with. It may seem like a distraction, but this speaks to the power and significance of SubNetX.

We appreciate the reviewer's suggestion and agree that a brief description of ARBRE and ATLASx would provide valuable context, helping readers understand the scale and significance of the data underpinning SubNetX.

To address this, we have added the following description after the first mention of these databases in the Results section:

"ARBRE is a highly curated database of balanced biochemical reactions with a particular focus on industrially relevant aromatic compounds, comprising approximately 400,000 reactions. ATLASx, on the other hand, is one of the largest networks of predicted biochemical reactions, containing over 5 million reactions that span a wide range of biochemical space. Together, these databases serve as a comprehensive foundation, enabling SubNetX to process large-scale reaction networks and extract meaningful pathways efficiently."

This addition underscores the power and scalability of SubNetX, highlighting its ability to handle the vast biochemical diversity represented in these databases.

6.) How were thermodynamic properties computed? What concentration ranges were considered in thermodynamic constraints? Were experimental concentration values used?

We integrated the following section into the Methods:

"Thermodynamic Curation of the Models

Incorporating thermodynamic information allows us to constrain reaction directionality according to the second law of thermodynamics, specifically by utilizing the Gibbs free energy of each reaction. The thermodynamic curation of the metabolic subnetworks predicted by SubNetX was carried out in two steps:

1. Gibbs Free Energy of Formation

We estimated the Gibbs free energy of formation for each compound, along with its associated estimation error. For each metabolite, we retrieved public identifiers (e.g., ChEBI, KEGG, SEED) and used these to obtain compound structures in SMILES format. Using Marvin (version 18.1, 2018, ChemAxon), the SMILES were converted to their primary protonation state at pH 7 and then to MDL Molfiles. These Molfiles were used with the Group Contribution Method (GCM) to estimate the standard Gibbs free energy of formation for each metabolite and the associated estimation error.

2. Gibbs Free Energy of Reactions

The Gibbs free energy of reactions within the subnetwork was calculated based on the corrected Gibbs free energies of compounds, which were adjusted for pH and ionic

strength using the Debye-Hückel approximation. For a reaction with m components, the Gibbs free energy, $\Delta_r G'$, was calculated as:

$$\Delta_r G' = \sum_{j=1}^m n_j \Delta_f G_j'^o + RT \ln \left(\prod_{j=1}^m x_j^{n_j} \right), \quad (7)$$

where:

- n_j : stoichiometric coefficient of compound j ,
- $\Delta_f G_j'^o$: standard Gibbs free energy of formation of compound j ,
- $x_j^{n_j}$: concentration of compound j ,
- R : ideal gas constant ($R = 8.31 \cdot 10^{-3} \frac{KJ}{K mol}$),
- T : temperature ($T = 298 K$).

In the absence of metabolomics data, generic concentration ranges were used. For intracellular metabolites (i.e., cytoplasmic or periplasmic), concentrations were constrained between 11 μ M and 50 mM, and for extracellular metabolites, between 10 nM and 100 mM.

7.) In mapping metabolites in ARBRE and ATLASx to host models in the methods, structure was used. Was stereochemistry considered? Generally, many cheminformatics-generated reactions are blind to stereochemistry, making union with metabolic models a challenge.

We thank the reviewer for raising this point. The ARBRE and ATLASx databases used in SubNetX are indeed blind to stereochemistry, a common limitation of cheminformatics-generated reaction networks. In our metabolite mapping, we used the first level of the InChIKey to unify reaction networks and models. This level encodes the 2D structure of compounds, facilitating consistent mapping across the datasets while ensuring compatibility with the SubNetX pipeline.

It is important to note that SubNetX is not restricted to 2D structures. Providing full InChIKeys, which include stereochemical information (3D structure), could address this limitation and enable the incorporation of stereochemical considerations into the analysis. By leveraging such additional data, SubNetX can further enhance the accuracy and specificity of pathway predictions. We have clarified this flexibility in the manuscript.

"A key feature of SubNetX is its reliance on the first level of InChIKey encoding, which represents the 2D molecular structure of compounds, to unify reaction networks and models. While this approach ensures compatibility across datasets and maintains consistency within the SubNetX framework, it does not account for stereochemistry. However, SubNetX is designed with flexibility to accommodate additional levels of molecular detail. By utilizing full InChIKeys that encode stereochemical data, future applications of SubNetX could incorporate 3D structural information, thereby addressing stereochemical considerations and enhancing the precision of pathway predictions. This capability allows for further refinement of metabolite mapping and reaction analysis, particularly in contexts where stereochemistry plays a critical role in biochemical processes."

8.) What kind of linear optimization solvers were used for the work? I would suspect open-source solvers would still not be up to solving many portions of the SubNetX workflow, but it would be good to report this.

We appreciate the reviewer's query about the linear optimization solvers used in this work. To address this, we have added the following information to the manuscript:

" The SubNetX code was tested on macOS and Linux systems. For this study, we used Mac Pro machines with 32 GB of RAM and a 3.7 GHz Quad-Core processor. Installation requires Python 3.6 with the following packages: networkx, rdkit, and cobra. The project's GitHub repository provides a detailed installation guide and all dependencies. The optimization problems were solved using the commercial solver ILOG CPLEX, version 12.8.0."

Reviewer #2 (Remarks on code availability):

The code is in a well-documented public git repository. There are example parameter files for all bioproducts studied in the paper. There is also a default file for users to try their own compounds. There are clear installation instructions as well, and the dependencies are very reasonable.

Reviewer #3 (Remarks to the Author):

The authors present a new method for combining the strengths of retrobiosynthesis and constraint-based pathway design. The paper is well written, and presents a useful method that would be broadly useful to the field. The work does support the conclusion, and open source software is provided to reproduce these methods. I thank the authors for putting in the effort to provide well documented code to allow others to leverage this method.

I have a few questions and suggestions for the authors:

1) What external data sources or software tools were required to do the full analysis presented here, and are any of them proprietary or commercial? For example, it mentions that BNICE.ch was integrated to include novel hypothetical reactions, but the code repository says the data source is ARBRE. It would be helpful to provide some information on what is required to use the code, such as system requirements or other software and data.

We thank the reviewer for their question and for highlighting the importance of clarifying the external data sources and software tools used in our analysis.

Our manuscript specifies that BNICE.ch was employed to extend the predicted reaction network for compounds beyond the scope of ARBRE (e.g., scopolamine and tadalafil). However, it is important to note that BNICE.ch itself is not integrated into the SubNetX pipeline. Instead, it was used to generate an extended reaction network that was then input into SubNetX. The BNICE.ch code is available to academics upon request, independent of the SubNetX implementation. A database of balanced biochemical reactions is required to use the SubNetX pipeline. For this study, ARBRE and ATLASx were used as primary data sources. SubNetX does not rely on proprietary software, ensuring it remains accessible to the research community.

We have updated the "Code and Data Availability" section of the manuscript to provide system requirements and dependencies as follows:

" The SubNetX code was tested on macOS and Linux systems. For this study, we used Mac Pro machines with 32 GB of RAM and a 3.7 GHz Quad-Core processor. Installation requires Python 3.6 with the following packages: networkx, rdkit, and cobra. The project's GitHub repository provides a detailed installation guide and all dependencies. No proprietary software is required to run the SubNetX pipeline. The optimization problems were solved using the commercial solver ILOG CPLEX, version 12.8.0."

2) To what extent does this differ methodologically, and compare performance wise to the previously published GEM-Path method, which integrates retrobiosynthesis with genome scale models (PubMed ID 25080239)?

We thank the reviewer for their insightful question, which allows us to highlight the methodological and performance differences between SubNetX and GEM-Path.

The key methodological distinction lies in how each tool approaches network expansion and pathway prediction within the metabolic capabilities of the host organism:

1. **GEM-Path**

GEM-Path employs an iterative process that tightly integrates retrobiosynthesis with genome-scale metabolic models (GEMs). It filters pathways early in the process, eliminating infeasible options based on the host's biochemical and biophysical constraints.

This targeted approach prevents the unnecessary expansion of the reaction network around the target compound and ensures that only pathways with a high likelihood of feasibility are retained. While this methodology is computationally efficient, it may limit the exploration of novel or less obvious pathways that could arise from broader network expansion.

2. SubNetX

In contrast, SubNetX performs a broader network expansion through retrobiosynthesis, connecting all cofactors and cosubstrates to the host GEM. Instead of filtering pathways during the expansion step, SubNetX incorporates the entire expanded subnetwork into the GEM. It then applies a Mixed-Integer Linear Programming (MILP) optimization framework to identify feasible pathways. This approach enables SubNetX to holistically analyze the expanded reaction network and uncover pathways that may not have been apparent during the initial retrosynthesis step. The broader exploration provided by SubNetX can reveal alternative or unconventional metabolic routes, which are particularly valuable in complex metabolic systems or microbial communities.

Performance Comparison

While both methods are effective, SubNetX's broader approach allows for a more comprehensive analysis of the metabolic landscape, particularly when aiming to identify non-obvious pathways. However, this broader exploration comes with higher computational requirements compared to the more streamlined and targeted process of GEM-Path. SubNetX's ability to integrate larger subnetworks is made feasible by leveraging robust MILP techniques and modern computational resources.

To emphasize this distinction, we have added the following clarification to the manuscript:

"SubNetX takes a broader, more integrative approach to identifying potential pathways than other pathway prediction methods that rely solely on retrobiosynthesis followed by host feasibility filtering. While narrowing the search space early by eliminating infeasible pathways can improve efficiency, it may also limit the discovery of novel or less obvious pathways. SubNetX expands the metabolic network through retrobiosynthesis by connecting cofactors and cosubstrates to the host GEM, then integrates the expanded subnetwork into the host model using Mixed-Integer Linear Programming. This comprehensive approach allows SubNetX to explore alternative pathways and uncover new opportunities for metabolic engineering, particularly in complex metabolic systems or microbial communities."

We hope this comparison clarifies the methodological and performance differences between SubNetX and GEM-Path.

3) Although it is impressive how well this works for the 8 compounds listed, it would be extremely valuable to perform this analysis on a large set of molecules, such as all FDA approved compounds, and get detailed aggregate statistics. 8 compounds is a small set and may not be very representative. If possible, a quantitative comparison to some existing widely used retrobiosynthesis tools would be valuable.

We thank the reviewer for their insightful suggestion. We agree that extending SubNetX to a larger set of molecules, such as all FDA-approved compounds, would provide valuable aggregate statistics and a more comprehensive evaluation of its performance. This is indeed a promising avenue for future studies. However, conducting such an extensive analysis is beyond the scope of the current work. While the SubNetX algorithm is computationally efficient, processing each compound still requires significant time. For instance, generating 100 alternative pathways per compound in this study took between 40 minutes and 8 hours on a 2013 Mac Pro with 32 GB of RAM and a 3.7 GHz quad-core processor, depending on the

complexity of the compound. Scaling this analysis to more than thousands of compounds would necessitate substantial computational resources, including parallelization and access to high-performance computing clusters, to handle the workload efficiently.

We acknowledge the importance of such analyses regarding a quantitative comparison with other retrosynthesis tools. However, it is important to note a key distinction in the design of SubNetX: its reliance on a curated input database that already includes the generation of novel reactions. This preprocessing step is crucial for the biosynthetic design, but it also means that the quality and scope of the input database heavily influence the performance of SubNetX.

Many existing retrosynthesis tools integrate reaction generation directly into their workflows, making direct comparisons challenging. SubNetX, by contrast, is designed to flexibly incorporate any database after proper preprocessing. This modularity ensures that it can adapt to advances in cheminformatics and database curation. Still, it also underscores that its results are not entirely comparable with tools that generate reactions dynamically.

We hope this response clarifies the scope of the current work and provides a perspective on future directions for expanding and benchmarking SubNetX.

Reviewer #3 (Remarks on code availability):

The code is freely available, and appears well documented, and well written. I did not attempt to install and run the code to test it myself.

Authors response to the reviewers' comments

We thank the reviewers for their insightful, constructive, and positive comments. We have highlighted below our detailed responses to the suggested improvements and changes. The changes are also marked in the manuscript.

Reviewer #1 (Remarks to the Author):

The authors addressed some of my issues, while there are still two major concerns need to be addressed:

1. I understand that large-scale testing consumes time and computational resources, but I still believe that additional case studies such as synthetic drugs and other categories of natural products are necessary.

We sincerely thank the reviewer for their interest in our work. Our initial goal was to present SubNetX as a flexible and customizable pipeline that allows users to design synthetic pathways for different compounds in different hosts. To demonstrate its applicability, we initially selected a limited set of compounds while making the pipeline and its instructions openly available, allowing users to generate pathways tailored to their specific goals.

In response to the reviewer's suggestion, we have now expanded our analysis to include 70 different natural and synthetic compounds (Supplementary Table 1). For each compound, we extracted the subnetwork, calculated the maximum theoretical yield, and identified the shortest biosynthetic pathway. This expanded dataset is now available to the community, providing a valuable resource for designing synthetic pathways for any of these compounds in *E. coli*.

It is important to note that our general conclusions remain unchanged. In particular, all 70 compounds can be produced at maximum theoretical yield, meaning that all carbon atoms from the substrate are preserved towards the target compound. We have updated the manuscript to reflect these additions.

2. None of the experimental pathways is recovered, this should be further discussed. Do all experimental pathways contain unbalanced reactions? In an ideal situation, SubNetX could predict pathways with reactions from any organism, including *E. coli*, so it “should” predict the reported pathways. By the way, there is no Supplementary Table 4 in the Supplementary Information.

We agree with the reviewer that the experimental pathways are not fully recovered. However, SubNetX successfully identified key segments of these pathways, suggesting that it captured the essential biochemistry surrounding the experimentally implemented routes. For example, for ajmalicin, the case discussed in the paper, the same intermediates were recovered in our pathways (7-deoxyloganate, loganate, secologanin, and strictosidine), although the exact reactions differed (the cofactors and cosubstrates were different). Notably, all the reactions

missing from the predicted subnetworks are either imbalanced or with unknown mechanism, which prevents their inclusion by design. We now clarify this point in the revised manuscript:

“It is noteworthy that although the exact reactions could be different, the predicted pathways passed through the same set of intermediates as the experimentally implemented pathways (e.g., 7-deoxyloganate, loganate, laganin, secologanin, and strictosidine to produce ajmalicin), suggesting that SubNetX captured the essential biochemistry surrounding the experimentally implemented pathways. The difference in reaction steps was mainly due to the difference in cofactors and cosubstrates.”

Regarding the Supplementary Table 4 (now named Supplementary Table 5), you can find it as a separate excel file provided in the submission package.

Reviewer #2 (Remarks to the Author):

The authors have responded rigorously to all of my prior comments, making many clarifying changes to the manuscript. As such, I have no new comments or suggestions at this time.

Reviewer #2 (Remarks on code availability):

No changes from prior comments.

Reviewer #3 (Remarks to the Author):

While the authors addressed most of my concerns, I still feel that 8 compounds is not a large enough set to systematically evaluate the ability of the method to generalize to diverse targets. The authors mention using an older computer system which can process a single compound between 40 minutes and 8 hours, which would allow for a test set in one week of between 21 and 252 compounds. My suggestion would be to run at least as many as can be run in a full week, with an attempt to maximize the chemical diversity of the test set in an unbiased way.

We thank the reviewer for their comment on how to improve our work. We have expanded our analysis to include 70 different compounds, while trying to maintain chemical diversity. For each compound, we extracted the subnetwork, calculated the maximum theoretical yield, and identified the shortest biosynthetic pathway. This expanded dataset is now available to the community, providing a resource for designing synthetic pathways for any of these compounds in *E. coli* (please see the response to the Reviewer #1 for further details). We have also updated the paper accordingly.

The authors have thoroughly addressed all of my other questions and concerns.

Reviewer #3 (Remarks on code availability):

The code is freely available, and appears well documented, and well written. I did not attempt

to install and run the code to test it myself.